# Phantasus, a web application for visual and interactive gene expression analysis

**Maksim Kleverov[1,2†], Daria Zenkova[1†], Vladislav Kamenev[1†], Margarita Sablina[1], Maxim N Artyomov[2], Alexey A Sergushichev[1,2]\***

[1]ITMO University, Computer Technologies Laboratory, Saint Petersburg, Russian Federation; [2]Washington University in St. Louis School of Medicine, Department of Pathology and Immunology, St Louis, United States

**Abstract** Transcriptomic profiling became a standard approach to quantify a cell state, which led to the accumulation of huge amount of public gene expression datasets. However, both reuse of these datasets or analysis of newly generated ones requires significant technical expertise. Here, we present Phantasus: a user-friendly web application for interactive gene expression analysis which provides a streamlined access to more than 96,000 public gene expression datasets, as well as allows analysis of user-uploaded datasets. Phantasus integrates an intuitive and highly interactive JavaScript-based heatmap interface with an ability to run sophisticated R-based analysis methods. Overall Phantasus allows users to go all the way from loading, normalizing, and filtering data to doing differential gene expression and downstream analysis. Phantasus can be accessed online at https://alserglab.wustl.edu/phantasus or can be installed locally from Bioconductor (https://bioconductor.org/packages/phantasus). Phantasus source code is available at https://github.com/ctlab/phantasus under an MIT license.

**\*For correspondence:**
alsergbox@gmail.com

†These authors contributed equally to this work

**Competing interest:** The authors declare that no competing interests exist.

## Editor's evaluation

This study introduces Phantasus, a useful tool accessible through both web and local applications, designed to analyze transcriptome data derived from microarray or RNA-seq technologies. The compelling tool facilitates normalization, data visualization, and differential expression analysis. Phantasus represents a valuable contribution to the biomedical community, enabling individuals without extensive bioinformatics expertise to analyze new transcriptomic data or reproduce studies effectively.

## Introduction

Transcriptomic profiling is a ubiquitous method for whole-genome-level profiling of biological samples (*Stark et al., 2019*). Moreover, the deposition of these data into one of the public repositories has become a standard in the field, leading to the accumulation of a huge amount of publicly available data. The most significant example is the NCBI Gene Expression Omnibus (GEO) project (*Barrett et al., 2013*), which stores information from more than 225,000 studies.

Sharing of transcriptomic data opens up possibilities for reusing them: instead of carrying out a costly experiment, a publicly available dataset can be used, thus decreasing the cost and accelerating the research (*Byrd et al., 2020*). However, the standard approach for gene expression analysis requires significant technical expertise. In particular, many analysis methods are implemented in R as a part of Bioconductor project ecosystem (*Gentleman et al., 2004*), and thus one has to have programming skills in R to use them. On the other hand, domain knowledge is beneficial to improve quality

control of the data, which is especially important when working with the publicly available data, as well as generation of biological hypotheses (*Wang et al., 2016*).

A number of applications have been developed with the aim to simplify analysis of transcriptomic datasets (see Appendix 1 for details). In particular, web-based applications remove the burden of setup and configuration from the end users, thus lowering the entry threshold. Shiny framework (*Chang, 2022*) revolutionized the field as it became easy to create a web interface for R-based pipe-lines, which led to a significant growth of web applications for gene expression analysis (*Ge et al., 2018*; *Hunt et al., 2022*; *Mahi et al., 2019*; *Nelson et al., 2017*). However, such applications gener-ally have limited interactivity due to mainly server-side computations. Shiny-independent applications can be more interactive, but they suffer from lack of native R support and require reimplementation of existing methods from scratch (*Gould, 2016*; *Alonso et al., 2015*).

Here, we present Phantasus: a web application for gene expression analysis that integrates highly interactive client-side JavaScript heatmap interface with an R-based back-end. Phantasus allows us to carry out all major steps of gene expression analysis pipeline: data loading, annotation, normal-ization, clustering, differential gene expression, and pathway analysis. Notably, Phantasus provides streamlined access to more than 96,000 microarray and RNA-seq datasets from the GEO database, simplifying their reanalysis. Phantasus can be accessed online at https://alserglab.wustl.edu/phan-tasus or can be installed locally from Bioconductor. Phantasus is open source, and its code is available at https://github.com/ctlab/phantasus under MIT license (*Sergushichev and Kamenev, 2024*).

## Results

### Phantasus web application

We developed a web application called Phantasus for interactive gene expression analysis. Phantasus integrates a JavaScript-rich heatmap-based user interface originated from Morpheus (*Gould, 2016*) with an R back-end via the OpenCPU framework (*Ooms, 2014*). The heatmap graphical interface provides an intuitive way to manipulate the data and metadata: directly in a web browser the user can create or modify annotations, edit color schemes, filter rows and columns, and so on. On the other hand, the R back-end provides a way to easily run a multitude of computational analysis methods available as R packages. Altogether, this architecture (*Figure 1*) ensures a smooth experience for performing all common analysis steps: loading datasets, normalization, exploration, visualization, differential expression and gene set enrichment analyses.

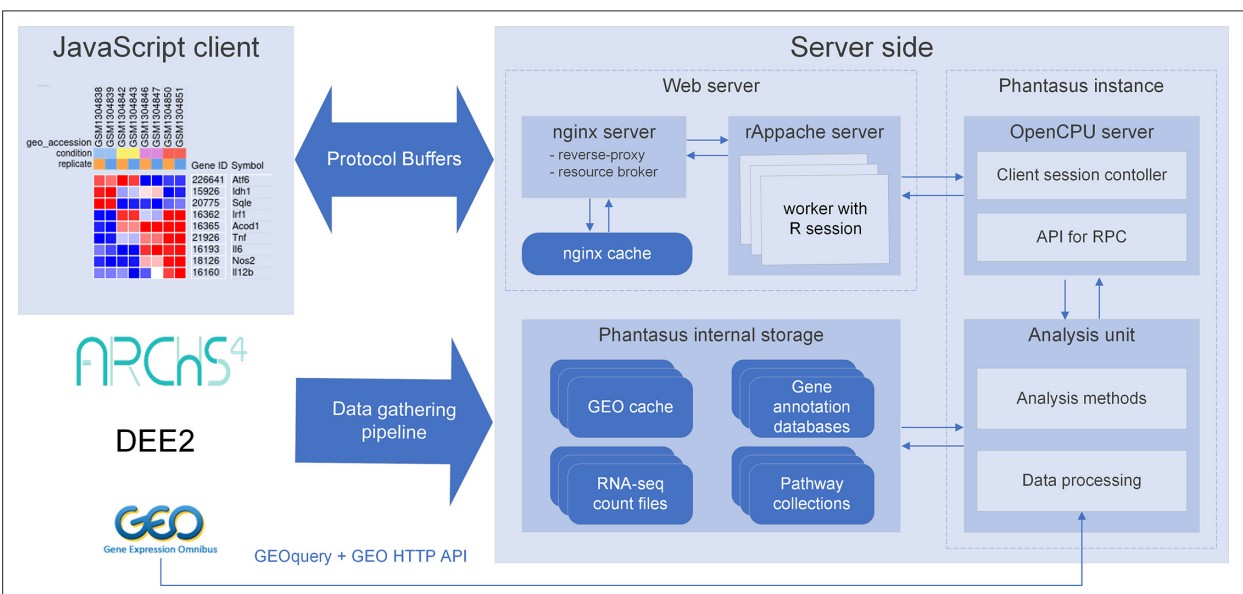

**Figure 1.** Overview of Phantasus architecture. The front-end interface is a JavaScript application that requests the web server to load the data and perform resource-consuming tasks. The core element of the back-end is the OpenCPU-based server, which triggers the execution of R-based analysis methods. Protocol Buffers are used for efficient client-server dataset synchronization.

Several options for loading the gene expression data into Phantasus are available. First, datasets from GEO (*Barrett et al., 2013*) can be loaded by their identifier. Phantasus supports microarray datasets, which are loaded directly from GEO, as well as RNA-sequencing datasets, for which counts data from third-party databases are used (see section 'Available datasets' for details). Second, datasets can be loaded from a gene expression table file in GCT, TSV, CSV, and XLSX formats (with a support for gzip-archived files). Finally, a set of curated datasets is available directly from the home page.

A number of methods can be used to prepare, normalize, and explore the gene expression table. In particular, it is possible to aggregate microarray probe-level data to gene levels, transform and filter the data, conduct a principal component analysis (PCA), perform k-means or hierarchical clustering, etc. These tools allow you to perform a thorough quality control of the dataset and remove outliers if present.

When the dataset is properly filtered and normalized, differential expression analysis using limma (*Ritchie et al., 2015*) or DESeq2 pipelines (*Love et al., 2014*) can be carried out. These results can then be used with other web services for downstream analysis, with shortcuts for pathway analysis with Enrichr (*Kuleshov et al., 2016*) and metabolic network analysis with Shiny GAM (*Sergushichev et al., 2016*). Additionally, gene set enrichment analysis (GSEA) can be done directly in Phantasus as implemented in the fgsea package (*Korotkevich et al., 2021*).

All the plots produced by Phantasus during the data exploration and analysis can be exported as vector images in SVG format. This includes heatmaps, PCA plots, gene profiles, enrichment plots, etc. The obtained images can be used for publications as is or adjusted in a vector graphics editor.

Another option for presenting final or intermediate results is session link sharing. When a link is generated, a snapshot with the current dataset and its representation – annotations, color scheme, sample dendrograms, etc. – are saved on the server. The link can be shared with other users, and, when opened, restores the session.

## Stand-alone Phantasus distribution

Aside from using the official mirror (https://alserglab.wustl.edu/phantasus), there is a possibility to set up Phantasus locally. Phantasus can be installed as an R package from Bioconductor (https://

| Source | Full support | Partial support | Total |
|---|---|---|---|
| NCBI GEO | 96649 | 40061 | 136710 |
| — GEO Microarray | 49846 | 27807 | 77653 |
| —— GEO curated annotation | 39621 | 1123 | 40744 |
| —— Parsed user-provided annotation | 10225 | 678 | 10903 |
| —— Other user-provided annotation | 0 | 26006 | 26006 |
| — GEO RNA-seq | 46803 | 12254 | 59057 |
| —— ARCHs4 | 35088 | 7543 | 42631 |
| ——— *Homo sapiens* | 16776 | 4347 | 21123 |
| ——— *Mus musculus* | 17031 | 2880 | 19911 |
| ——— Other organisms | 1281 | 316 | 1597 |
| —— DEE2 (not in ARCHs4) | 11715 | 4711 | 16426 |
| ——— *Homo sapiens* | 3556 | 2196 | 5752 |
| ——— *Mus musculus* | 3507 | 1340 | 4847 |
| ——— Other organisms | 4652 | 1175 | 5827 |

**Figure 2.** Dataset availability in Phantasus. For fully supported datasets, gene expression data is accompanied by gene annotations in a standardized format. Partial support datasets have either incomplete gene expression matrix or gene annotations.

The online version of this article includes the following source data for figure 2:

**Source data 1.** Source file for table shown in *Figure 2*.

bioconductor.org/packages/phantasus) or loaded as a Docker image (https://hub.docker.com/r/alser-glab/phantasus). In both cases, almost all of the Phantasus functions will be available from the start.

Some of the Phantasus features require additional server-side setup. Extended support of GEO datasets requires preprocessed expression matrices and platform annotations. Identifier mapping requires organism annotation databases. Pathways enrichment requires pathway databases. The R package and Docker image versions of Phantasus can automatically handle this setup by downloading the required files from https://alserglab.wustl.edu/files/phantasus/minimal-cache. The detailed instructions for installing Phantasus locally are available at https://ctlab.github.io/phantasus-doc/installation.

Importantly, we have introduced a highly scalable data service (HSDS) server (https://alserglab.wustl.edu/hsds/?domain=/counts,), facilitating access to HDF5 files containing precomputed gene count matrices from the official Phantasus mirror. Through this service and a helper R package phantasusLite (https://bioconductor.org/packages/phantasusLite), a stand-alone Phantasus instance selectively loads the count matrix exclusive to the specified dataset, avoiding the need to burden local installations with unnecessarily large files. However, for users prioritizing increased reactivity, the option remains to load these files from the aforementioned cache mirror.

An important feature of the stand-alone version of Phantasus is the ability to share manually curated datasets. Similar to Phantasus session link sharing, one can generate a named session consisting of a dataset and its visual representation. Link to this named session (e.g. https://alserglab.wustl.edu/phantasus/?preloaded=GSE53986.Ctrl_vs_LPS) can then be shared for the other users to view. Such a predictable display of the data can be particularly useful in a publication context.

## Available datasets

Phantasus provides streamlined access to more than 96,649 GEO datasets. For these datasets, the expression values and gene identifiers (Entrez, ENSEMBL, or Gene Symbol) are readily available (*Figure 2*). Moreover, these datasets are used to populate the initial Phantasus cache, and thus they have low loading times.

Of these 96,649 datasets, 49,846 are microarrays based on 2767 platforms. For 1347 platforms, GEO databases have machine-readable annotations in the *annot.gz* format with Entrez gene and Gene symbol columns, corresponding to 39,621 datasets. The remaining 10,225 datasets are obtained from platforms that do not have a GEO-provided annotation. For these 1420 platforms, we have automatically marked up user-provided annotations in *SOFT* format to extract gene identifiers and convert the annotations into *annot.gz* format.

The RNA-seq subset of the datasets with streamlined access consists of 46,803 datasets. As GEO does not store expression values for RNA-seq datasets, we rely on other databases for the expression data. The first-priority database for RNA-seq gene counts is ARCHS4 (Human, Mouse, and Zoo

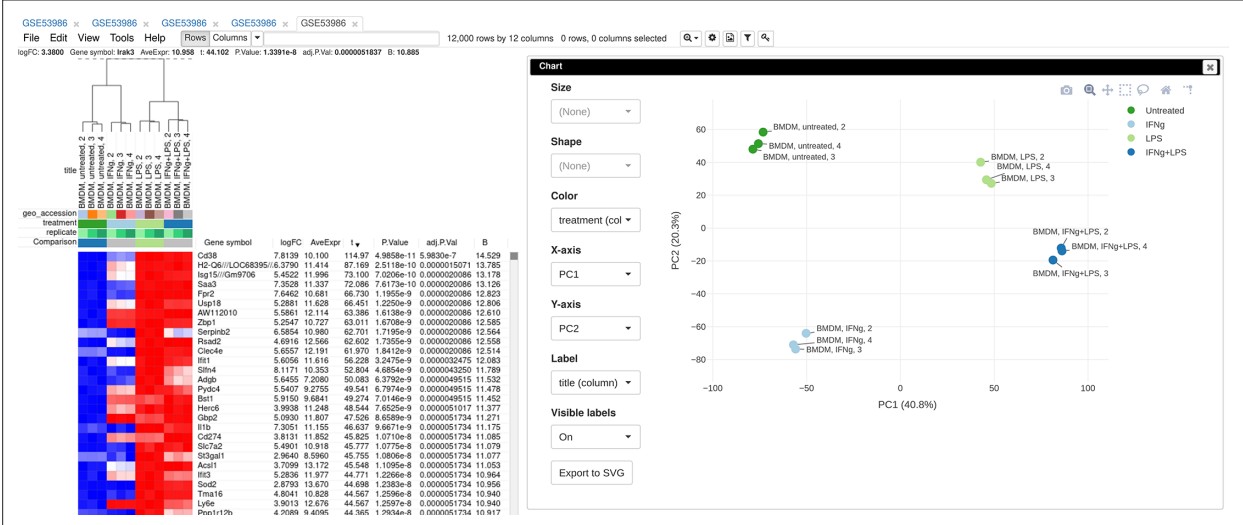

**Figure 3.** Example of analyzed dataset GSE53986 with normalized gene expression values, filtered outliers, hierarchically clustered columns, and rows annotated with differential expression analysis between untreated and LPS-treated macrophages.

versions), which covers 35,088 datasets. The other source is the DEE2 database (human, mouse, and other available organisms), which covers an additional 11,715 datasets. The DEE2 database contains run-level quantification, so it has been preprocessed to sum read counts into sample-level tables.

## Case study 1: Basic usage

To illustrate the basic usage of Phantasus, we will consider the dataset GSE53986 (*Noubade et al., 2014*) from the GEO database (*Figure 3*). This dataset consists of 16 samples of bone marrow-derived macrophages, untreated and treated with three stimuli: LPS, IFNg, and combined LPS + IFNg. The gene expression was measured with Affymetrix Mouse Genome 430 2.0 Array. Here, we give an overview of the steps; the full walk-through for the analysis is available in Appendix 2.

As the first step of the analysis, the dataset can be loaded and normalized. The dataset is loaded straightforwardly by the *GSE53986* identifier. Because this is a microarray dataset, internally the gene expression values are obtained from GEO. In this particular case, the expression values have not been normalized, but it can be done in Phantasus. From the available normalization options, we select log2 scaling and quantile normalization. Furthermore, we can aggregate microarray probe-level expression values into gene-level expression. We chose the *Maximum Median Probe* method, which retains only a single probe per gene, the one that has the highest median expression value. Finally, we can filter out lowly expressed genes, for example, by keeping only the top 12000 expressed genes.

After the normalization step, we can apply a number of exploratory techniques. In particular, we can do a PCA, k-means gene clustering, and hierarchical clustering of the samples. From these analyses, we can discover that there is an overall good concordance between the replicates of the same treatment, with an exception of the first replicate in each group. We can conclude that these samples are outliers and remove them before the downstream analysis.

Finally, we can do a comparison between the sample groups, for example, by comparing untreated and LPS-treated samples. As the data has been normalized, we can apply *limma* for differential gene expression analysis. The result appears as additional gene annotation columns: p-values, log-fold-changes, and other statistics. Next, we can use differential expression results for a pathway enrichment analysis: for example, we can use R-based GSEA via the *fgsea* package or we can use external tools, such as Enrichr.

## Case study 2: Data reanalysis

To highlight Phantasus's ability to reanalyze publicly available data in a context of a biological study, let us consider a study by *Mowel et al., 2017*. The study considers a genomic locus *Rroid* linked by the authors to homeostasis and function of group 1 innate lymphoid cells (ILC1). The authors hypothesized that *Rroid* locus controls ILC1s by promoting the expression of Id2 gene, a known regulator of ILCs. To confirm this hypothesis, the authors generated an Id2-dependent gene signature based on an existing transcriptomic data (*Shih et al., 2016*) and showed its deregulation in *Rroid*-deficient cells.

The computational analysis described above linking Rroid and Id2 can be replicated in Phantasus in a straightforward way (see Appendix 3 for the detailed walk-through).

First, we can open GEO dataset GSE76466 (*Shih et al., 2016*), containing gene expression data for Id2-deficient NK cells. Notably, GSE76466 is an RNA-sequencing dataset, without gene expression values stored directly in the GEO database; however, Phantasus loads the dataset leveraging precomputed expression values from the ARCHS4 project (*Lachmann et al., 2018*). Then we can perform differential gene expression analysis with the limma tool, comparing Id2-deficient and wild-type (WT) natural killer (NK) cells. Id2-dependent gene signature can be obtained by sorting the genes by *t* column.

Second, RNA-sequencing dataset GSE101458 (*Mowel et al., 2017*), generated by Mowel and colleagues for Rroid-deficient NK cells, can also be opened in Phantasus. There we can do differential gene expression analysis with DESeq2 and remove lowly expressed genes. Finally, we can input the generated Id2-dependent gene signature into Phantasus gene search field and use GSEA plot tool to obtain an enrichment plot, similar to the one presented by Mowel and colleagues, confirming a potential regulation via Id2.

## Discussion

Here, we present Phantasus: a web tool designed to simplify the analysis of gene expression data and provide streamlined access to tens of thousands of publicly available datasets. This is an important area of development, as evidenced by the numerous tools addressing this and similar objectives. In Appendix 1, we compare some of these tools, and we anticipate the development of more of them in the future. However, this diversity is highly beneficial to the broad research community as it enhances the overall accessibility of gene expression analysis by allowing researchers to find the most suitable tool for their needs.

Phantasus's uniqueness lies in its highly interactive heatmap-based user interface, integrated with an R and Bioconductor-based back-end. In our experience, this combination effectively supports users from both computational and biological backgrounds. It also provides avenues for extending support to analysis methods available within the R and Bioconductor ecosystem.

A major focus of Phantasus's development was to provide easy access to publicly available datasets from the GEO database. To our knowledge, Phantasus is the only web application for gene expression analysis that offers streamlined access to both microarray and RNA-seq datasets. Currently, it combines approximately 96,000 datasets, with additional datasets available following some manipulations by the user. About half of these datasets are RNA-sequencing datasets, where quantified gene expression values are not stored in GEO. For this purpose, we integrated the ARCHS4 and DEE2 databases and also made them available remotely from the R environment via the phantasusLite package in a GEOquery-compatible manner.

Phantasus is readily available online at https://alserglab.wustl.edu/phantasus, but it can also be installed as a Bioconductor R package at https://bioconductor.org/packages/phantasus. For further convenience, we provide a Docker image at https://hub.docker.com/r/alserglab/phantasus. Phantasus documentation, including comprehensive installation instructions, is available at https://ctlab.github.io/phantasus-doc/. The Phantasus source code is available at https://github.com/ctlab/phantasus under an MIT license (*Sergushichev and Kamenev, 2024*).

## Materials and methods
### Web application architecture

Phantasus is a web application that combines an interactive graphical user interface with access to a variety of R-based analysis methods (*Figure 1*). The front-end, which is JavaScript-based, derives from the Morpheus web application designed for matrix visualization and analysis (*Gould, 2016*). The back-end is written in R, with an OpenCPU server (*Ooms, 2014*) translating HTTP-queries from the client into R procedure calls.

The JavaScript client is responsible for the matrix visualization, as well as certain analysis methods. In particular, steps like subsetting the dataset, working with annotations, and basic matrix modification (e.g., log-transformation, scaling, etc.) have client-side implementation. Furthermore, the client supports additional visualization methods such as row profile plots, volcano plot, and others.

The analysis methods that require external data or algorithms are implemented in the form of the `phantasus` R package to be carried out on the server side. The operations include differential gene expression analysis, PCA, pathway analysis, and others. Commonly, these methods rely on functions that are already available in the existing R packages; for such methods, only wrapper R functions are implemented.

The OpenCPU server is a core component of the Phantasus back-end. The server provides an HTTP API for calling computational methods implemented in R. For each call, OpenCPU creates a new R environment with the required data, in which the method is then executed. OpenCPU can manage these R environments both in a standard single-user R session and, with the help of rApache, in a multi-user manner inside an Apache web server.

The transfer of large objects between the server and the client exploits a binary Protobuf protocol. The Phantasus back-end uses the protolite R package (*Ooms, 2021*) for object serialization and deserialization. The front-end relies on the protobuf.js module (*Coe, 2020*).

For further performance improvement, Nginx server is used to wrap the OpenCPU server. Nginx server caches the results of the OpenCPU method calls. If the same method with the same data is

called again, the cached result can be returned without any additional computations. Furthermore, Nginx is used to serve static content and manage permissions.

## Data sources and data gathering

The main data source for Phantasus is the NCBI GEO database (*Barrett et al., 2013*). All of the GEO datasets are identified by a GSEnnnnn accession number (with a subset of the datasets having an additional GDSnnnnn identifier). However, depending on the type of dataset, the processing procedure is different.

The majority of gene expression datasets in the GEO database can be divided into two groups: microarray data and RNA-seq data. While the experiment metadata is available for all of the datasets, the expression matrices are provided only for the microarray datasets. Phantasus relies on the GEOquery package (*Davis and Meltzer, 2007*) to load the experiment metadata (for all datasets) and expression matrices (for microarray datasets) from GEO.

When a GEO RNA-seq dataset is requested by the user, Phantasus refers to precomputed gene counts databases available in the internal storage. In particular, data from ARCHS4 (*Lachmann et al., 2018*) and DEE2 (*Ziemann et al., 2019*) projects are used. Both of these projects contain gene counts and metadata for RNA-seq samples related to different model organisms including but not limited to mouse and human. For any requested RNA-seq dataset, the gene counts are loaded from a single database, whichever covers the highest number of samples.

Next, Phantasus stores gene annotation databases that are used to map genes between different identifier types. These databases are stored in SQLite format compatible with the *AnnotationDbi* R package (*Pagès, 2022*). Currently, only human and mouse databases are available, which are based on `org.Hs.eg.db` and `org.Mm.eg.db` R packages, respectively.

Pathway databases are stored to be used for GSEA. Currently, gene set collections include the GO biological processes database (*Ashburner et al., 2000*), the Reactome database (*Gillespie et al., 2022*), and the MSigDB Hallmark database (*Liberzon et al., 2011*) for human and mouse.

Finally, for faster access, Phantasus dataset cache is automatically populated by a large compendium of datasets. The automatic pipeline is Snakemake-based and consists of four steps.

First, the pipeline converts DEE2 files into ARCHs4-like HDF5 files. During this procedure, the expression values of runs provided by DEE2 are summed up to the sample level. The second step checks for which microarray platforms GEO contains a curated machine-readable annotation in *annot.gz* format. The third step tries to generate the machine-readable annotation for the rest of the microarray platforms from the annotations available in the *SOFT* format. Currently, this step has produced an additional 1300 *annot.gz* files. The last step goes over all of the microarray datasets with a machine-readable annotation and over all of the RNA-seq datasets with the counts available in ARCHS4 or DEE2. For each such dataset, the cached entry with all of the data and metadata is created and stored.

A snapshot of Phantasus internal storage is available at https://alserglab.wustl.edu/files/phantasus/minimal-cache. It contains preprocessed count files, automatically marked-up annotations, and gene and pathways databases. This snapshot can be used for a local Phantasus setup.

## Acknowledgements

The project was supported by the Ministry of Science and Higher Education of the Russian Federation (Priority 2030 Federal Academic Leadership Program).

---

## Additional information

### Funding

| Funder | Grant reference number | Author |
| --- | --- | --- |
| Ministry of Science and Higher Education of the Russian Federation | Priority 2030 Federal Academic Leadership Program | Maksim Kleverov<br>Alexey A Sergushichev |

| Funder | Grant reference number | Author |
|---|---|---|

The funders had no role in study design, data collection and interpretation, or the decision to submit the work for publication.

## Author contributions

Maksim Kleverov, Software, Writing – original draft, Data curation, Writing – review and editing; Daria Zenkova, Vladislav Kamenev, Software, Writing – original draft; Margarita Sablina, Software; Maxim N Artyomov, Conceptualization, Methodology; Alexey A Sergushichev, Conceptualization, Software, Supervision, Funding acquisition, Methodology, Writing – original draft

## Author ORCIDs

Maksim Kleverov (i) http://orcid.org/0009-0008-0353-8945
Alexey A Sergushichev (i) http://orcid.org/0000-0003-1159-7220

## Decision letter and Author response

Decision letter https://doi.org/10.7554/eLife.85722.sa1
Author response https://doi.org/10.7554/eLife.85722.sa2

# Additional files

## Supplementary files

• MDAR checklist

## Data availability

The current article is a computational study, so no data have been generated. The application source code is available at https://github.com/ctlab/phantasus under an MIT licence. Previously published datasets were taken from the Gene Expression Omnibus; GEO series GSE53986 (*Noubade et al., 2014*); GEO series GSE76466 (*Delconte et al., 2016*); GEO series GSE101458 (*Mowel et al., 2017*).

The following previously published datasets were used:

| Author(s) | Year | Dataset title | Dataset URL | Database and Identifier |
|---|---|---|---|---|
| Noubade R, Wong K, Ota N, Rutz S, Eidenschenk C, Ding J, Valdez PA, Peng I, Sebrell A, Caplazi P, DeVoss J, Soriano RH, Modrusan Z, Hackney JA, Sai T, Ouyang W | 2014 | NRROS negatively regulates ROS in phagocytes during host defense and autoimmunity | https://www.ncbi.nlm.nih.gov/geo/query/acc.cgi?acc=GSE53986 | NCBI Gene Expression Omnibus, GSE53986 |
| Delconte RB, Shi W, Belz GT, Carotta S, Huntington ND | 2016 | The helix-loop-helix protein ID2 governs NK cell fate by tuning their sensitivity to interleukin-15 | https://www.ncbi.nlm.nih.gov/geo/query/acc.cgi?acc=GSE76466 | NCBI Gene Expression Omnibus, GSE76466 |
| Mowel WK, McCright SJ, Kotzin JJ, Collet M, Uyar A, Chen X, DeLaney A, Spencer SP, Virtue AT, Yang E, Villarino A, Kurachi M, Dunagin MC, Harms Pritchard G, Stein J, Hughes C, Fonseca-Pereira D, Veiga-Fernandes H, Raj A, Kambayashi T, Brodsky IE, O'Shea JJ, Wherry EJ, Goff LA, Rinn JL, Williams A, Flavell RA, Henao-Mejia J | 2018 | ILC1 lineage identity is determined by a cis-regulatory element marked by a novel lncRNA | https://www.ncbi.nlm.nih.gov/geo/query/acc.cgi?acc=GSE101459 | NCBI Gene Expression Omnibus, GSE101459 |

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

# Appendix 1

In this section, we consider existing software platforms for gene expression analysis. While they share common principles, the details of implementation vary, which affects their usability. For the comparison, we considered three aspects: (1) support for gene expression analysis steps, (2) data availability, and (3) user experience.

The general gene expression analysis workflow can be roughly divided into three stages: data preprocessing, exploratory analysis, and differential gene expression with pathway analysis. *Appendix 1—table 1* contains an overview of support of the key analysis steps implemented in the considered tools. At the data preprocessing stage, many tools allow us to load unnormalized gene expression matrices and provide a variety of matrix normalization approaches such as log-transformation, quantile normalization, scaling, and others. After the dataset is normalized, some of these tools provide instruments for dataset exploration like PCA and row/column clustering. Gene expression analysis usually ends by determining differentially expressed genes and pathway analysis to explain the observed difference. While the differential expression is available almost everywhere, the pathway analysis can be carried out only by about a half of the tools. In this half, the most notable tools are Gene Pattern, Babelomics, iDep, EXPANDER, and GEOexplorer, which support all the considered steps.

**Appendix 1—table 1.** Gene expression analysis key steps.

| Application | Normalization | Principal component analysis | Clustering | Differential expression | Pathway analysis |
|---|---|---|---|---|---|
| Gene pattern *Reich et al., 2006* | + | + | + | + | + |
| GEO2R *Barrett et al., 2013* | - | - | - | + | - |
| BicOverlapper2 *Santamaría et al., 2014* | - | - | - | + | + |
| Babelomics *Alonso et al., 2015* | + | + | + | + | + |
| Morpheus *Gould, 2016* | + | - | + | - | - |
| START *Nelson et al., 2017* | + | + | - | + | - |
| BioJupies *Torre et al., 2018* | + | + | + | + | + |
| iDEP *Ge et al., 2018* | + | + | + | + | + |
| Degust *Powell, 2019* | - | + | - | + | - |
| GREIN *Mahi et al., 2019* | + | + | - | + | - |
| EXPANDER *Hait et al., 2019* | + | + | + | + | + |
| RaNa-seq *Prieto and Barrios, 2019* | - | + | - | + | + |
| RNAdetector *La Ferlita et al., 2021* | - | - | - | + | - |
| GEOexplorer *Hunt et al., 2022* | + | + | + | + | + |
| Phantasus | + | + | + | + | + |

The huge difference between tools appears in the supported data sources and the availability of datasets (*Appendix 1—table 2*). In addition to a simple loading of gene expression matrices in tabular formats, which is a common practice, few tools allow us to load publicly available datasets from external sources, in particular from GEO database. However, depending on the type of the GEO dataset, microarray or RNA-seq, their support varies among the considered tools. The GEO database does not store gene expression matrices for RNA-seq datasets, thus tools like GREIN and RaNa-seq incorporate alignment and quantification of the raw data into their pipeline. Alternatively, precomputed count matrices from other projects, such as ARHCs4 (*Lachmann et al., 2018*) and DEE2 (*Ziemann et al., 2019*), can be used, as implemented in BioJupies and iDep.

**Appendix 1—table 2.** Gene expression sources.

| Application | User-provided data | GEO microarray | GEO RNA-seq |
|---|---|---|---|
| Gene pattern *Reich et al., 2006* | + | + | - |
| GEO2R *Barrett et al., 2013* | - | + | - |
| BicOverlapper2 *Santamaría et al., 2014* | + | + | - |
| Babelomics *Alonso et al., 2015* | + | - | - |
| Morpheus *Gould, 2016* | + | - | - |
| START *Nelson et al., 2017* | + | - | - |
| BioJupies *Torre et al., 2018* | + | - | +/- * |
| iDEP *Ge et al., 2018* | + | - | +/-* |
| Degust *Powell, 2019* | + | - | - |
| EXPANDER *Hait et al., 2019* | + | - | - |
| GREIN *Mahi et al., 2019* | - | - | + |
| RaNa-seq *Prieto and Barrios, 2019* | + | - | + |
| RNAdetector *La Ferlita et al., 2021* | + | - | - |
| GEOexplorer *Hunt et al., 2022* | - | + | - |
| Phantasus | + | + | +/-* |

*As processed in ARCHs4 and/or Dee2 projects.

Finally, we consider features that influence user experience (*Appendix 1—table 3*). There are three major types of architectures used in gene expression analysis applications, each with different trade-offs in reactivity, user setup, and development ease: Shiny-based web applications, non-Shiny web applications, and stand-alone applications.

Some non-Shiny web applications, such as GenePattern, BioJupies, and RaNa-seq, allow users to share sessions, enhancing reproducibility and facilitating collaborative work or publication. While many tools offer interactive gene expression heatmaps and other plots for data visualization and exploration, the support for working with metadata is often limited. Only four other tools (Babelomics, Morpheus, EXPANDER, RNAdetector) allow the editing of genes and sample annotations, and among them, only Babelomics and iDEP can map gene annotations using external gene annotation databases.

**Appendix 1—table 3.** User experience features.

| Application | Architecture | Saved sessions | Interactive heatmap | Interactive plots | Editing sample annotations | Editing gene annotations |
|---|---|---|---|---|---|---|
| Gene pattern *Reich et al., 2006* | Web non-Shiny | + | - | - | - | - |
| GEO2R *Barrett et al., 2013* | Web non-Shiny | - | - | - | - | - |
| BicOverlapper2 *Santamaría et al., 2014* | Local installation | - | +/-* | - | - | - |
| Babelomics *Alonso et al., 2015* | Web non-Shiny | - | - | + | + | + |
| Morpheus *Gould, 2016* | Web non-Shiny | +/- | + | + | + | - |
| START *Nelson et al., 2017* | Web Shiny | - | +/-* | + | - | - |
| BioJupies *Torre et al., 2018* | Web non-Shiny | + | +/-* | + | - | - |
| iDEP *Ge et al., 2018* | Web Shiny | - | +/-* | + | - | + |
| Degust *Powell, 2019* | Web non-Shiny | - | + | + | - | - |
| GREIN *Mahi et al., 2019* | Web Shiny | - | + | + | - | - |

*Appendix 1—table 3 Continued on next page*

*Appendix 1—table 3 Continued*

| Application | Architecture | Saved sessions | Interactive heatmap | Interactive plots | Editing sample annotations | Editing gene annotations |
|---|---|---|---|---|---|---|
| EXPANDER *Hait et al., 2019* | Local installation | - | + | + | + | - |
| RaNa-seq *Prieto and Barrios, 2019* | Web non-Shiny | + | +/-* | + | - | - |
| RNAdetector *La Ferlita et al., 2021* | Local installation | - | - | - | + | - |
| GEOexplorer *Hunt et al., 2022* | Web Shiny | - | + | + | - | - |
| Phantasus | Web non-Shiny | + | + | + | + | + |

*Limited number of genes.

Phantasus is a Shiny-independent web application capable of executing all key steps in gene expression analysis. It grants access to an extensive collection of publicly available datasets, encompassing both microarray and RNA-seq data. Through an interactive heatmap-based interface, users can explore data, modify annotations, create plots, etc. Furthermore, Phantasus delivers results in a publication-ready format, allowing users to export images to vector-based graphics formats and share session links. These links retain all user-defined settings, including colors, annotations, and filter options.

## Appendix 2

### Introduction

In this section, we show an example usage of Phantasus for the analysis of public gene expression data from the GEO database. It starts from loading data, normalization and filtering outliers, to doing differential gene expression analysis and downstream analysis.

To illustrate the usage of Phantasus, let us consider the public dataset from the GEO database GSE53986. This dataset contains data from experiments, where bone marrow-derived macrophages were treated with three stimuli: LPS, IFNg, and combined LPS + IFNg.

### Opening Phantasus

The simplest way to try Phantasus application is to go to the website https://alserglab.wustl.edu/phantasus where the latest versions are deployed. Alternatively, Phantasus can be installed locally (see section 'Stand-alone Phantasus distribution').

When Phantasus opens, the starting screen (*Appendix 2—figure 1*) should appear.

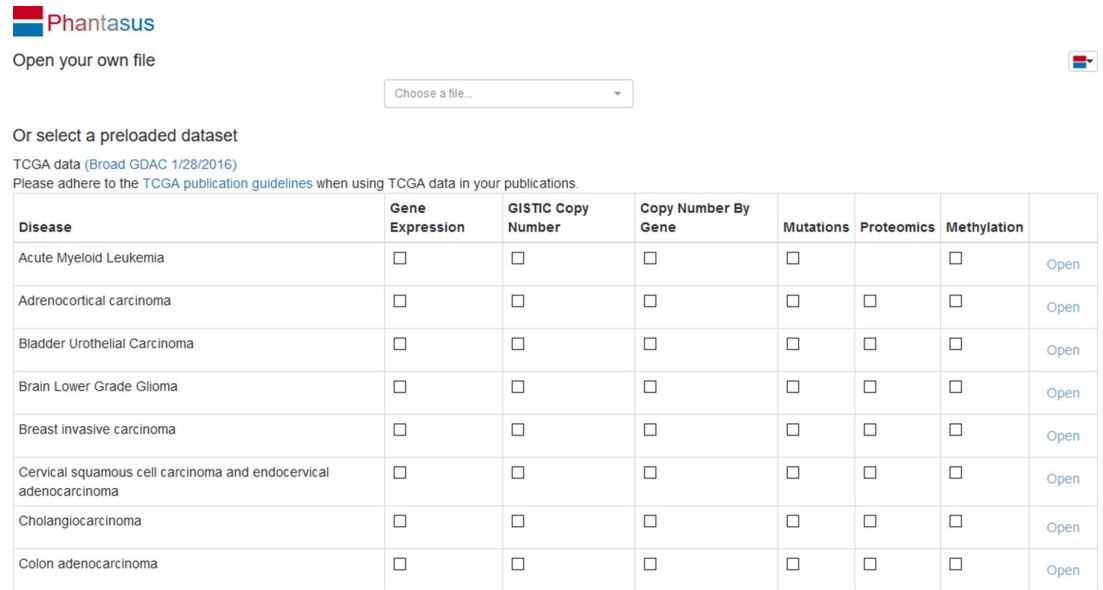

**Appendix 2—figure 1.** Phantasus starting screen.

### Preparing the dataset for analysis

Opening the dataset

Let us open the dataset. To do this, select the *GEO Datasets* option in *Choose a file …* dropdown menu. There, a text field will appear where GSE53986 should be entered. Clicking the *Load* button (or pressing *Enter* on the keyboard) will start the loading. After a few seconds, the corresponding heatmap (*Appendix 2—figure 2*) should appear.

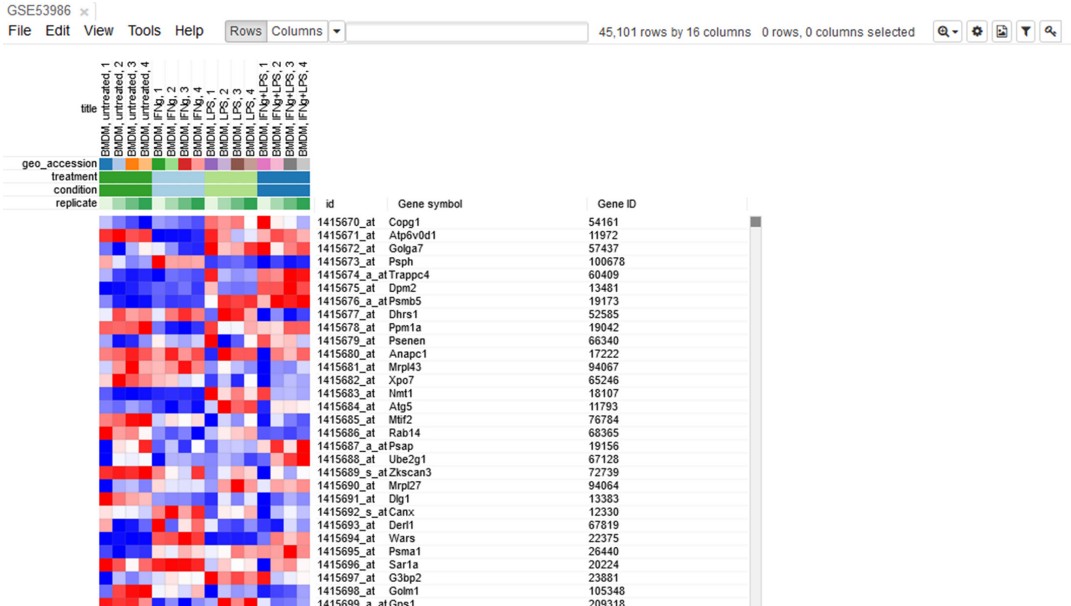

**Appendix 2—figure 2.** Heatmap of the loaded dataset.

On the heatmap, the rows correspond to genes (or microarray probes). The rows are annotated with *Gene symbol* and *Gene ID* annotations (as loaded from the GEO database). Columns correspond to samples. They are annotated with titles, GEO sample accession identifiers, and treatment field. The annotations, such as treatment, are loaded from user-submitted GEO annotations (they can be seen, for example, in *Characteristics* section at https://www.ncbi.nlm.nih.gov/geo/query/acc.cgi?acc=GSM1304836). We note that not for all of the datasets in GEO such proper annotations are supplied.

## Adjusting expression values

By hovering at a heatmap cell, gene expression values can be viewed. The large values there (*Appendix 2—figure 3*) indicate that the data is not log-scaled, which is important for most types of gene expression analysis.

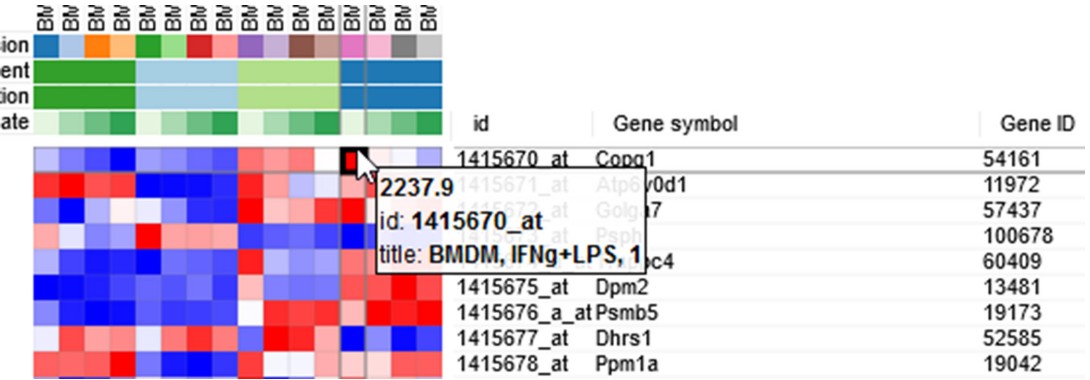

**Appendix 2—figure 3.** Heatmap cell tooltip.

For proper further analysis, it is recommended to normalize the matrix. To normalize values, go to *Tools/Adjust* menu and check *Log 2* and *Quantile normalize* adjustments (*Appendix 2—figure 4*).

**Appendix 2—figure 4.** Adjust tool.

A new tab with adjusted values will appear. All operations that modify gene expression matrix (such as adjustment, subsetting, and several others) create a new tab. This allows the user to revert the operation by going back to one of the previous tabs.

## Removing duplicate genes

Since the dataset is obtained with a microarray, a single gene can be represented by several probes. This can be seen, for example, by sorting rows by *Gene symbol* column (one click on column header), entering Actb in the search field and going to the first match by clicking the down-arrow next to the field. There are five probes corresponding to Actb gene in the considered microarray (*Appendix 2—figure 5*).

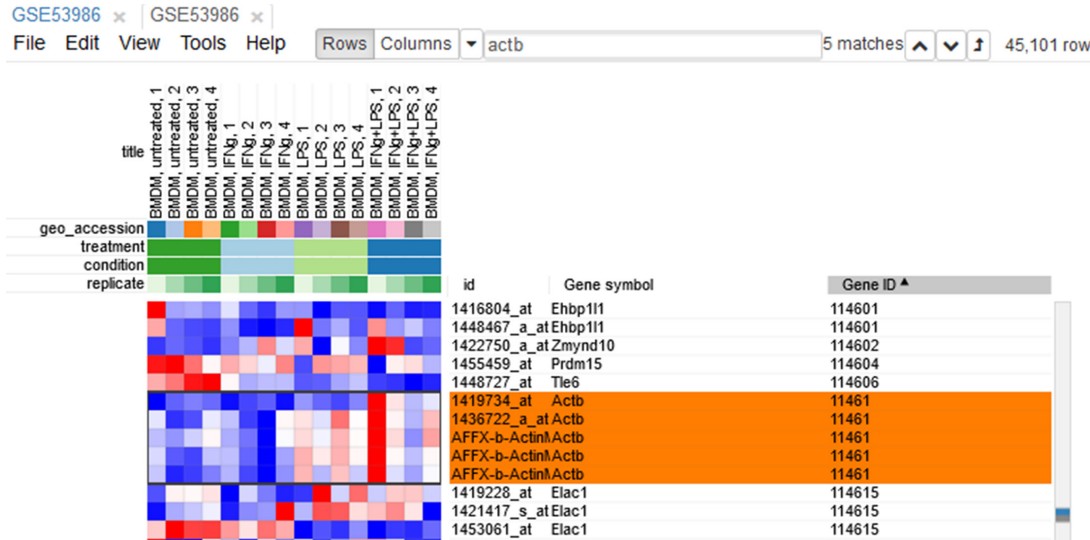

**Appendix 2—figure 5.** Actb duplicates.

To simplify the analysis, it is better to have one row per gene in the gene expression matrix. One of the easiest ways is to choose only one row that has the maximal median level of expression across all samples. Such a method removes the noise of lowly expressed probes. Go to *Tools/Collapse* and choose *Maximum Median Probe* as the method and *Gene ID* as the collapse field (**Appendix 2—figure 6**).

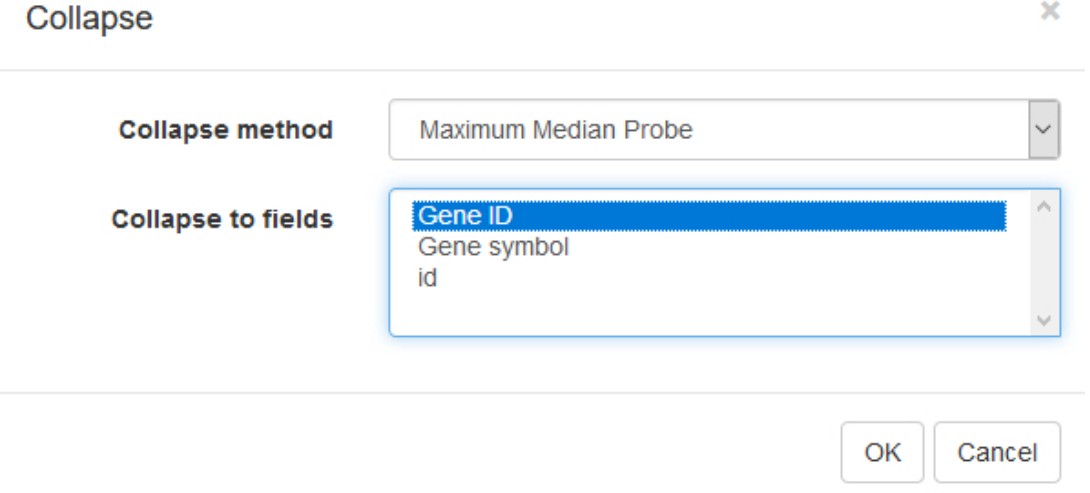

**Appendix 2—figure 6.** Collapse tool settings.

The result will be shown in a new tab.

## Filtering lowly expressed genes

Additionally, genes expressed at low levels can be explicitly filtered. It helps reduce noise and increase the power of downstream analysis methods.

First, we calculate mean expression of each gene using the *Tools/Create Calculated Annotation* menu. Select Mean operation, optionally enter a name for the resulting column (**Appendix 2—figure 7**), and click *OK*. The result will appear as an additional column in row annotations (**Appendix 2—figure 8**).

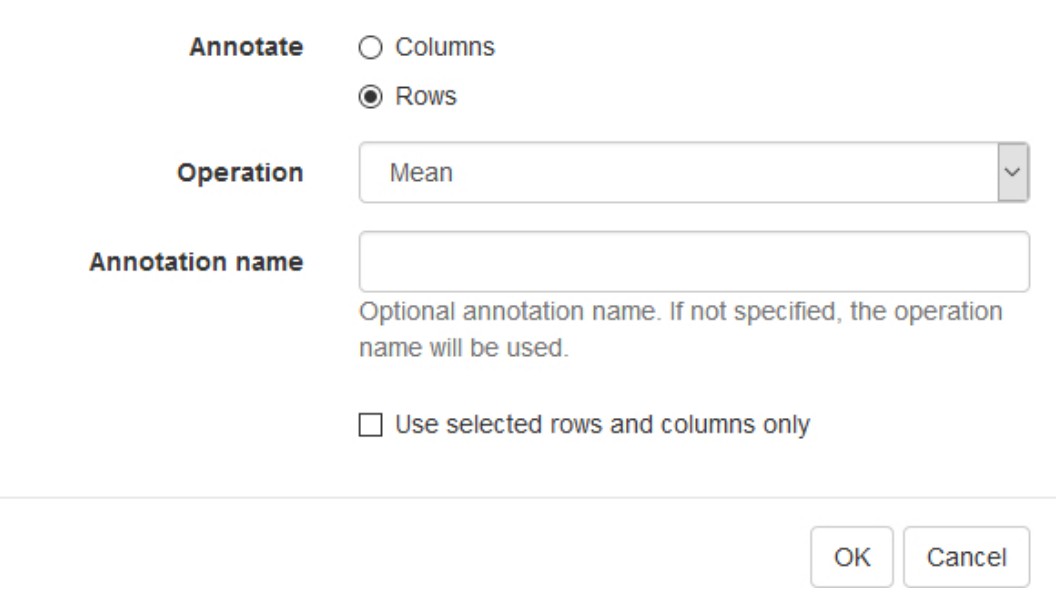

**Appendix 2—figure 7.** Row mean calculated annotation settings.

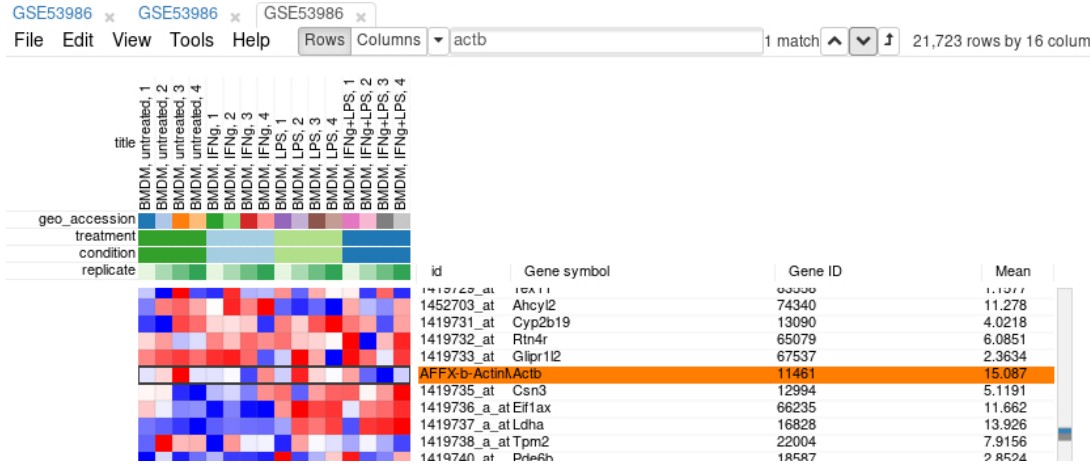

**Appendix 2—figure 8.** Heatmap with loaded row mean annotation.

Now this annotation can be used to filter genes. Open *Tools/Filter* menu. Click *Add* to add a new filter. Choose mean_expression as a *Field* for filtering. Then press *Switch to top filter* button and input the number of genes to keep. A good choice for a typical mammalian dataset is to keep around 10–12, 000 most expressed genes (*Appendix 2—figure 9*). Filter is applied automatically, so after closing the dialog with *Close* button only the genes passing the filter will be displayed (*Appendix 2—figure 10*).

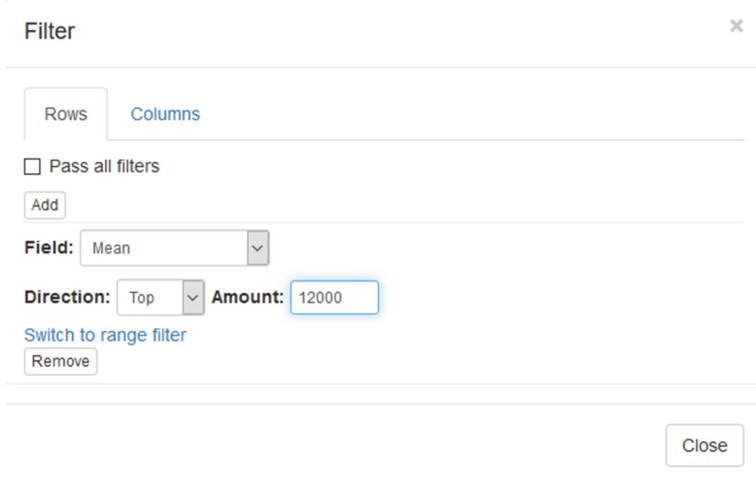

**Appendix 2—figure 9.** Filter tool settings.

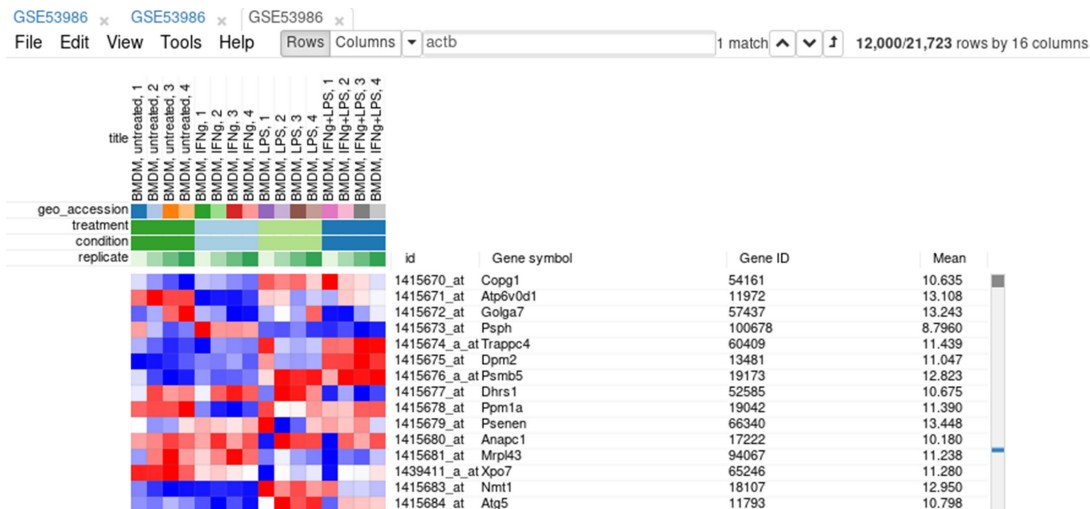

**Appendix 2—figure 10.** Heatmap with filtered genes.

Extracting these genes into a new tab is more convenient. For this, select all genes (click on any gene and press *Ctrl + A*) and use the *Tools/New Heat Map* menu (or press *Ctrl + X*).

Now you have the tab with a fully prepared dataset for further analysis. To easily distinguish it from other tabs, you can rename it by right clicking on the tab and choosing the *Rename* option. Let us rename it to `GSE53986_norm`.

It is also useful to save the current result to be able to return to it later. In order to save it, use the *File/Save Dataset* menu. Enter an appropriate file name (e.g., `GSE53986_norm`) and press OK. A file in text GCT format will be downloaded.

## Exploring the dataset
### PCA plot
One of the ways to assess the quality of the dataset is to use PCA method. This can be done using the *Tools/Plots/PCA Plot* menu (*Appendix 2—figure 11*).

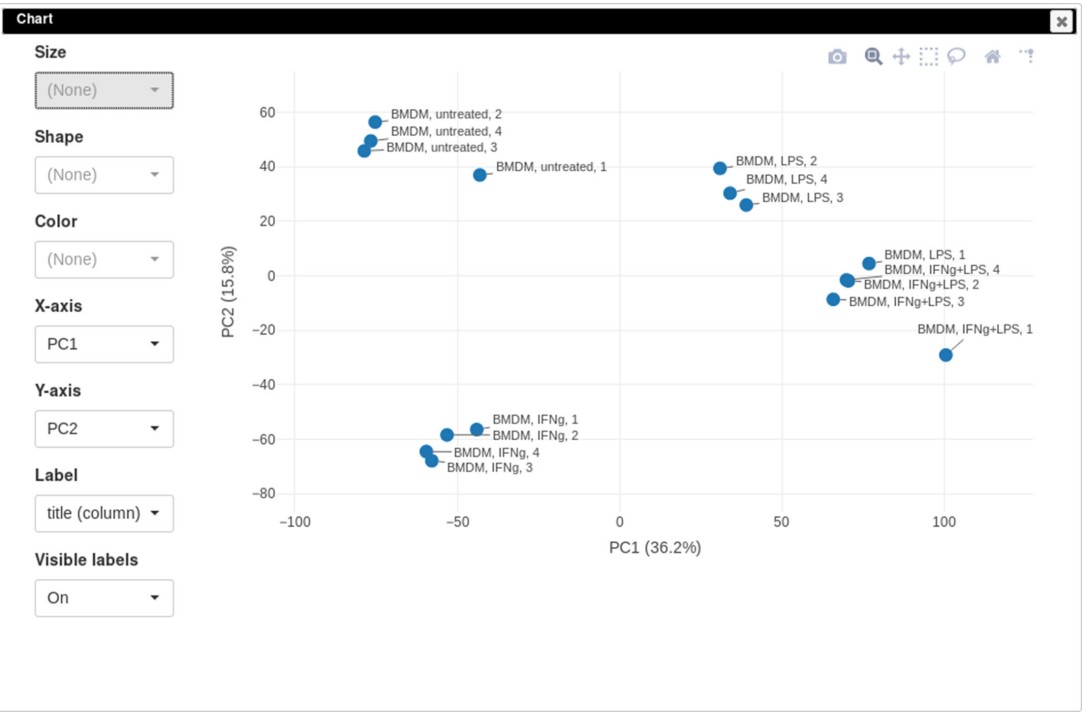

**Appendix 2—figure 11.** Principal component analysis (PCA) plot for dataset GSE53986.

You can customize color, size, and labels of points on the chart using values from annotation (*Appendix 2—figure 12*). Here we set color to come from *treatment* annotation.

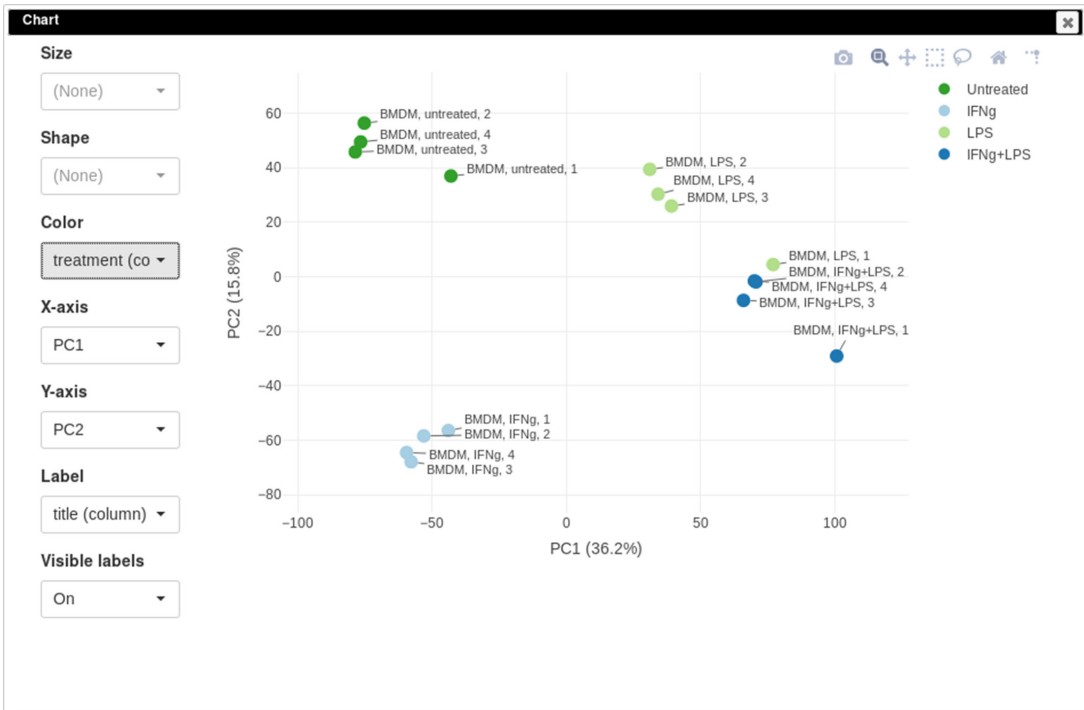

**Appendix 2—figure 12.** Customized principal component analysis (PCA) plot for the dataset GSE56986.

It can be seen that in this dataset the first replicates in each condition are outliers.

## k-means clustering

Another useful dataset exploration tool is k-means clustering. Use *Tools/Clustering/k-means* to cluster genes into 16 clusters (*Appendix 2—figure 13*).

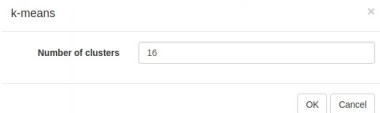

**Appendix 2—figure 13.** k-means tool settings.

Afterward, rows can be sorted by the *clusters* column. By using the menu *View/Fit to window*, one can get a 'bird's-eye view' on the dataset (*Appendix 2—figure 14*). Here also one can clearly see the outlying samples.

**Appendix 2—figure 14.** Clusters obtained using the k-means algorithm.

## Hierarchical clustering

*Tool/Hierarchical clustering* menu can be used to cluster samples and highlight outliers (and concordance of other samples) even further (*Appendix 2—figure 15*).

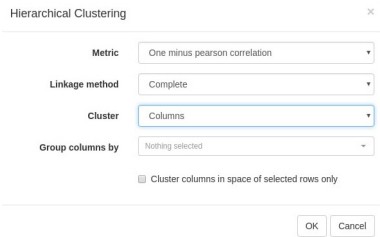

**Appendix 2—figure 15.** Hierarchical clustering settings.

## Filtering outliers

Now, when outliers are confirmed and easily viewed with the dendrogram from the previous step, you can select the good samples (*Appendix 2—figure 16*) and extract them into another heatmap (by clicking *Tools/New Heat Map* or pressing *Ctrl + X*).

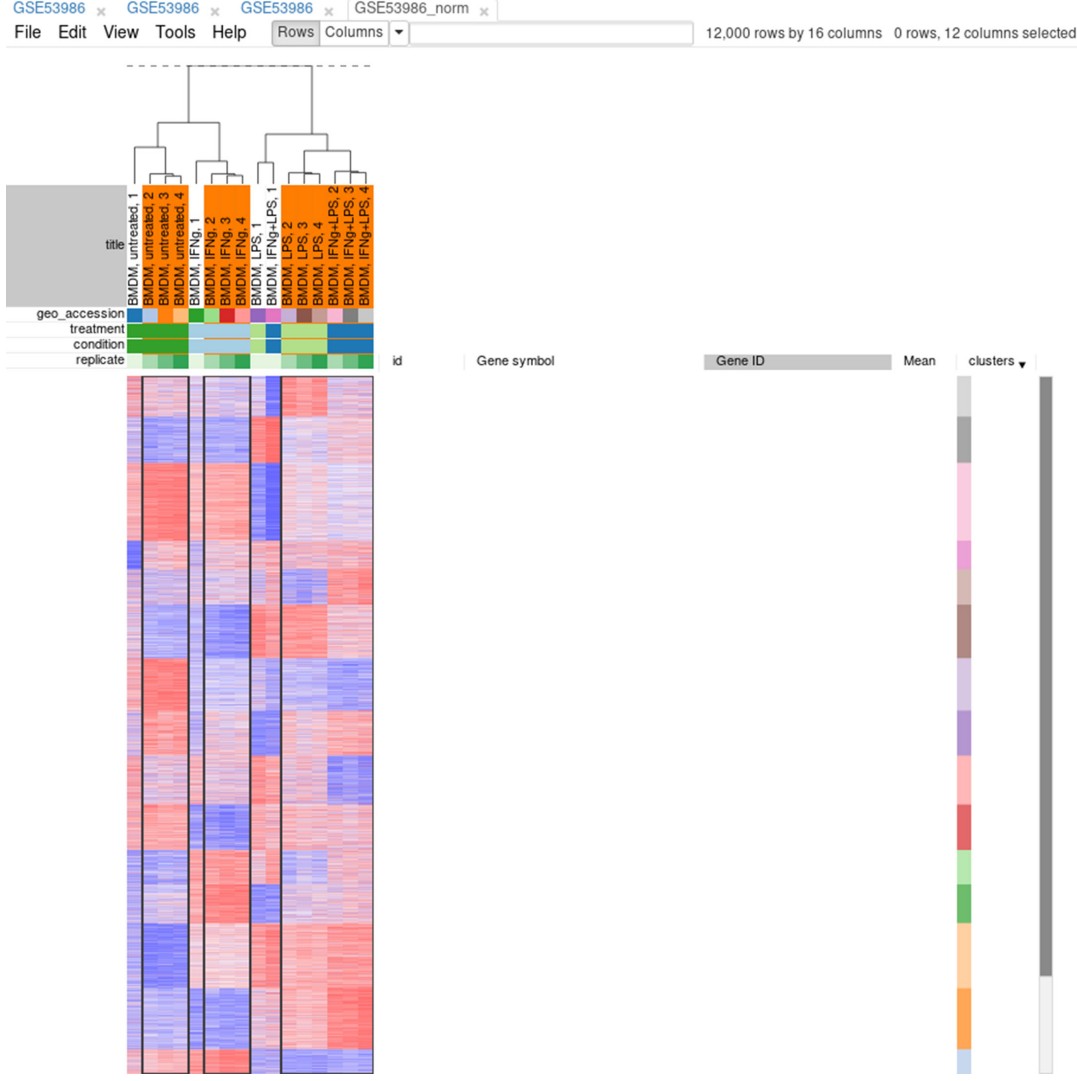

**Appendix 2—figure 16.** Good samples selection.

## Differential gene expression

### Applying *limma* tool

Differential gene expression analysis can be carried out with the *Tool/Differential Expression/limma* menu. Choose *treatment* as a *Field*, with *Untreated* and *LPS* as classes (***Appendix 2—figure 17***). Clicking *OK* will call the differential gene expression analysis method with the *limma* R package.

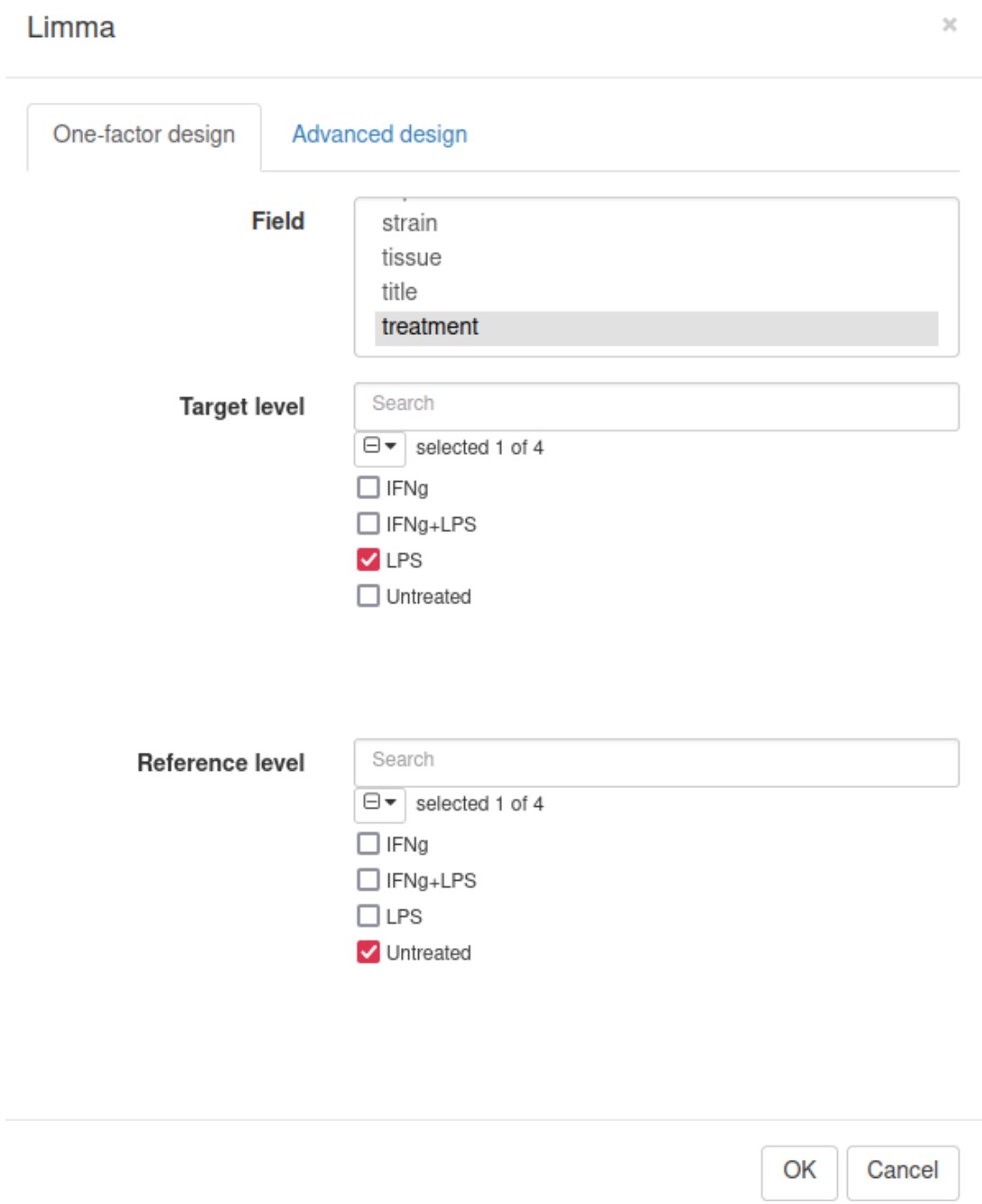

**Appendix 2—figure 17.** Settings for differential expression analysis by limma package.

The rows can be ordered by decreasing the *t*-statistic column to see which genes are the most upregulated upon LPS treatment (***Appendix 2—figure 18***).

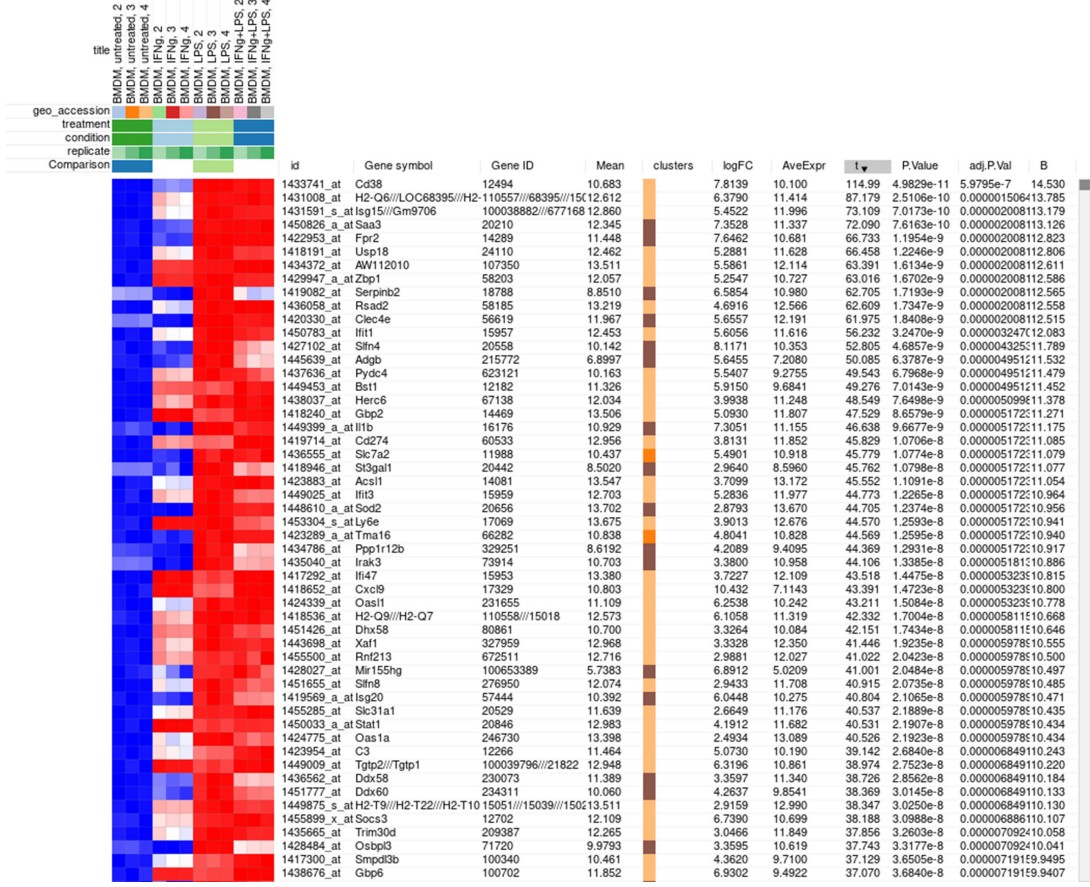

**Appendix 2—figure 18.** Differential expression analysis results for the dataset GSE53986.

## Pathway analysis with FGSEA

The results of differential gene expression can be used for pathway enrichment analysis with the *FGSEA* tool.

Open *Tools/Pathway Analysis/Perform FGSEA*, then select Pathway database, which corresponds to the specimen used in the dataset (*Mus musculus* in this example), ranking column, and column with ENTREZID or Gene IDs (*Appendix 2—figure 19*).

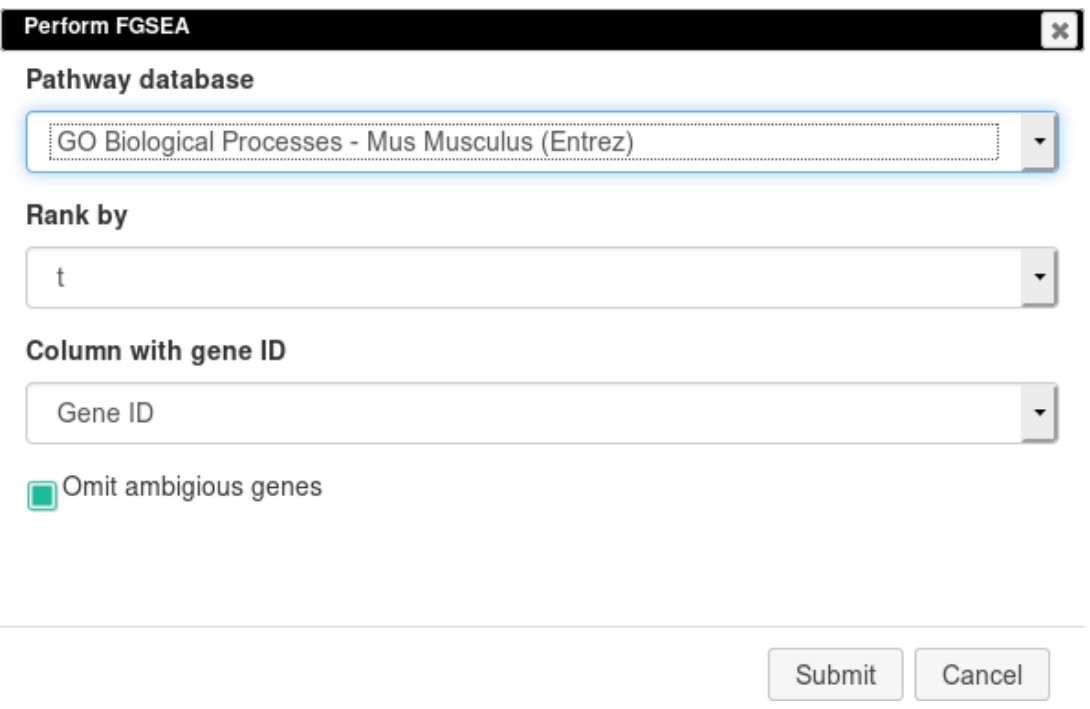

**Appendix 2—figure 19.** FGSEA settings.

Clicking Submit will open a new tab with pathways table (*Appendix 2—figure 20*).
Clicking on the table row will provide additional information on pathway: pathway name, genes in pathway, leading edge. You can save the result of the analysis in TSV format.

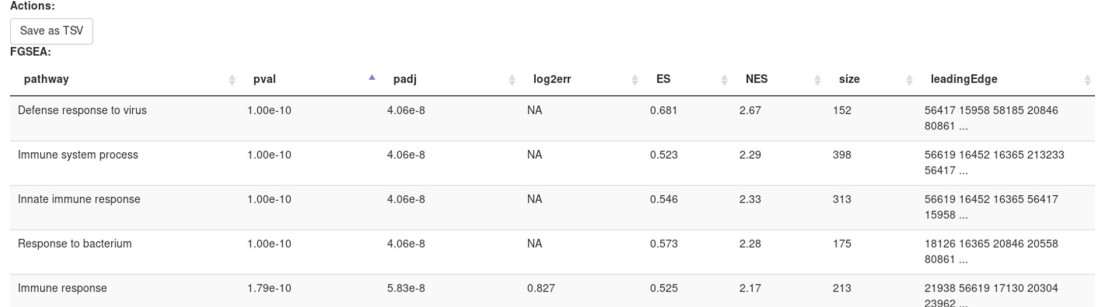

**Appendix 2—figure 20.** Pathways enriched by the FGSE tool for the dataset GSE54986.

# Appendix 3

## Overview

This section covers the replication of the study by *Mowel et al., 2017*. In the study, the authors show that *Rroid* locus control innate lymphoid cells (ILCs) by promoting *Id2* gene expression. Here we consider two stages of the analysis that can be performed using Phantasus:

- Obtaining *Id2*-dependent gene signature based on the dataset GSE76466
- Establishing *Rroid*-mediated regulation of the Id2-signature in the dataset GSE101458

Throughout this section, we will provide screenshots to demonstrate control settings for replication.

## Obtaining Id2-dependent gene signature

In the original paper, the authors use GSE76466 (*Shih et al., 2016*) dataset that contains WT and *Id2*-null NK cells samples to determine *Id2*-dependent signature. Using Phantasus, this dataset can be explored in a straightforward manner.

As the first step of the analysis, we will prepare the dataset. The dataset can be opened directly in Phantasus by its GEO identifier. For all the further steps, we will keep only WT and *Id2*-null samples. Select these samples and use the *Tools/New Heat Map* menu to create a new heatmap (*Appendix 3—figure 1*).

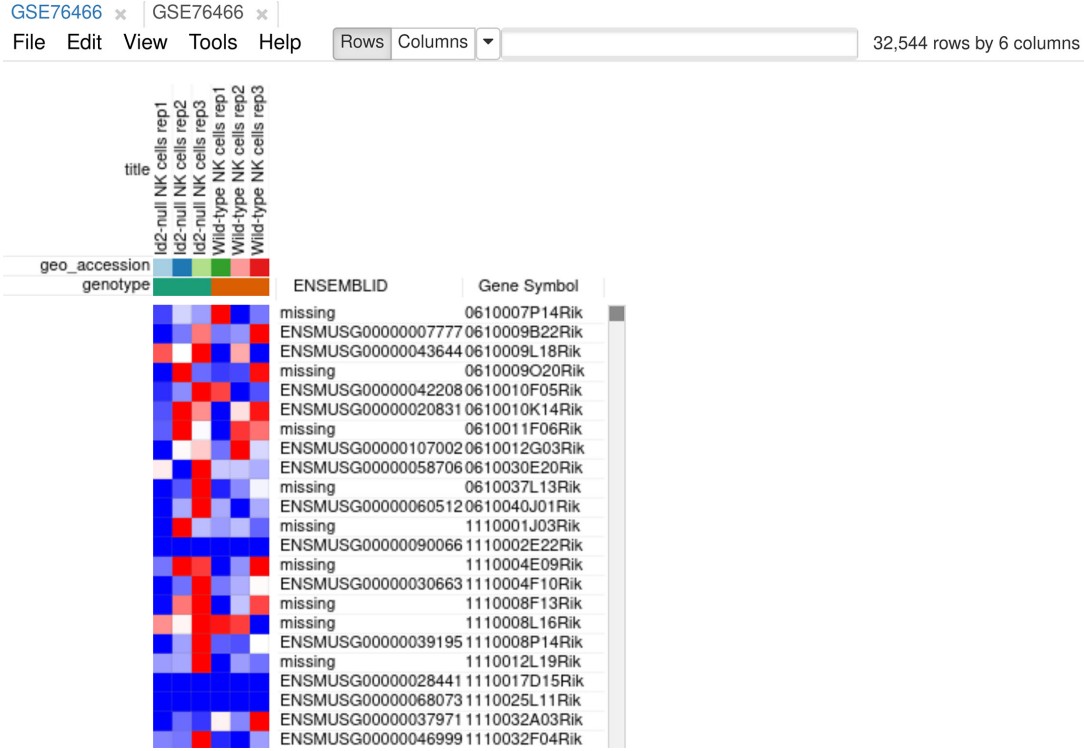

**Appendix 3—figure 1.** Heatmap for the dataset GSE76466 after samples filtration.

In the appeared tab, we need to normalize gene counts and filter out genes with low expression. With the help of *Tools/Advanced normalization/Voom* menu, it can be done in one step. Choose genotype as the factor of interest to build the design matrix for Voom and check automatic filtration option (*Appendix 3—figure 2*). Voom (*Law et al., 2014*) normalization converts expression counts into log2-counts per million values and will keep about 10,000 genes for further visualization and analysis.

**Voom: Mean-variance modelling at the observational level** ✕

Use column annotations to modify design matrix. By default all samples are treated as replicates.

**Factor** [ genotype ▾ ] ✕

[ add ]

Show design matrix

☑ Automatically filter out lowly expressed genes

[ OK ]  [ Cancel ]

**Appendix 3—figure 2.** Settings for the Voom tool.

To explore the normalized dataset, click on the *Tools/Plots/PCA plot* menu. The obtained PCA plot (*Appendix 3—figure 3*) shows clear separation between the WT and *Id2*-null samples. This encourages us to do next step and determine the genes that are differentially expressed between these two genotypes.

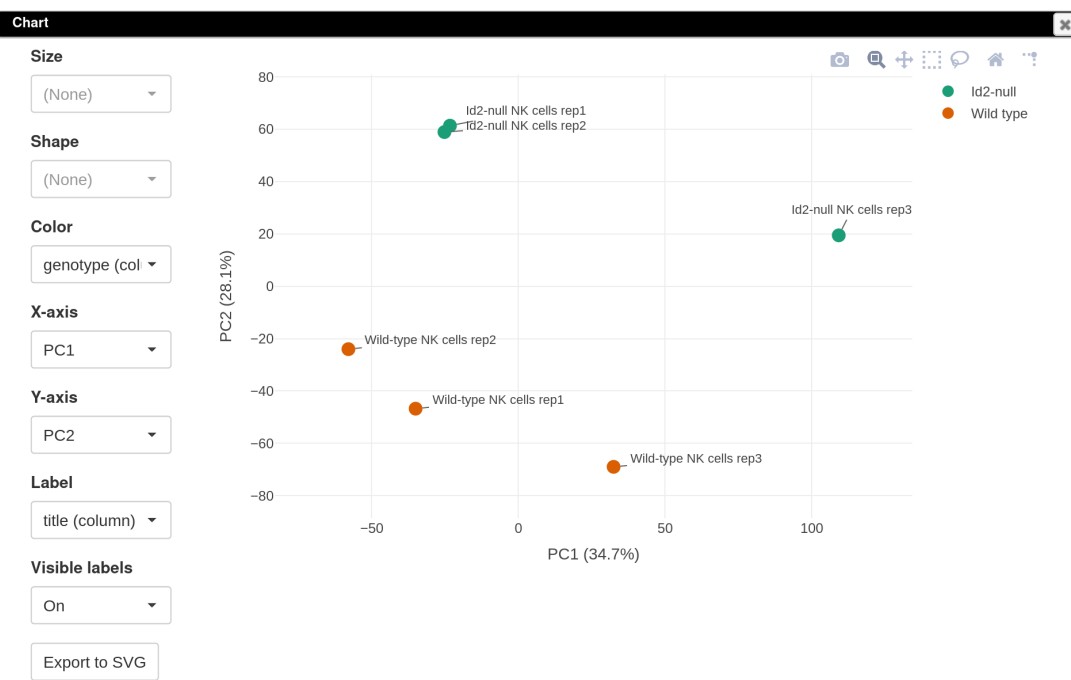

**Appendix 3—figure 3.** Principal component analysis (PCA) plot for the dataset GSE76466.

We will use limma tool (*Ritchie et al., 2015*) for the differential gene expression analysis. It will use the previously obtained weights from the Voom normalization to correct for mean–variance relation in RNA-seq data. In the *Tool/Differential Expression/Limma* menu, set *Id2*-null samples as the target level and WT as the reference (*Appendix 3—figure 4*).

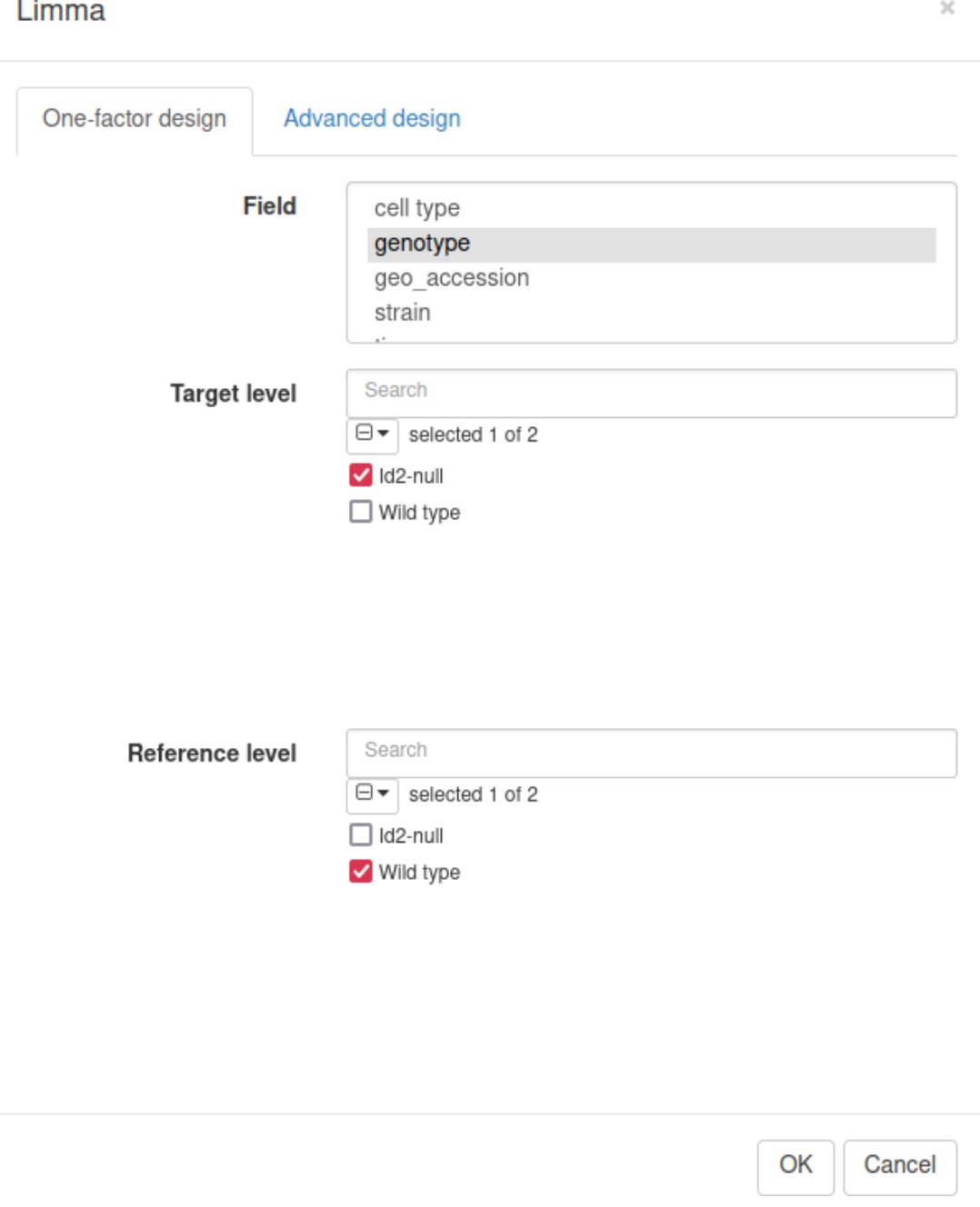

**Appendix 3—figure 4.** Limma tool settings for the dataset GSE76466.

A volcano plot generated using the *Tools/Plots/Volcano Plot* menu gives an overview of differential expression (*Appendix 3—figure 5*). In order to obtain the same picture, sort rows by *adj.P.Val* column, select the first few rows, and check the *Label by selected* option to add point annotations for the plot. Notably, *Id2* mRNA expression is downregulated in the knockout samples, which serves as a good positive control.

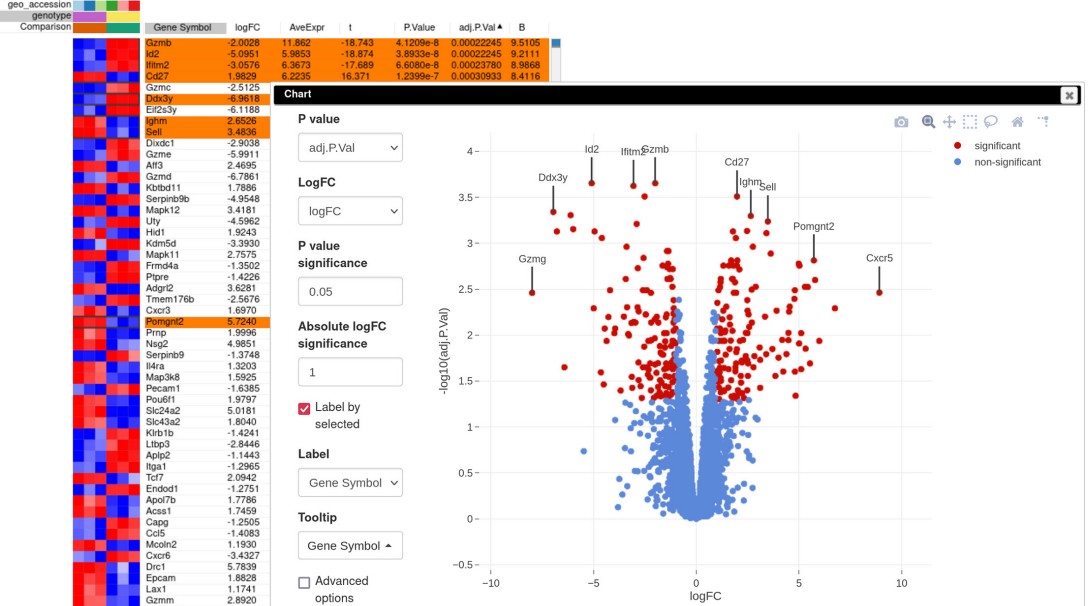

**Appendix 3—figure 5.** Volcano plot with differential gene expression results between wild-type (WT) and *Id2*-deficient samples for dataset GSE76466.

Desirable *Id2*-dependent gene signature, used by the authors, consists of the genes upregulated in *Id2*-null samples compared to WT. To obtain this signature, we can apply adj.P.Val < 0.05 and *t* ≥ 0 filters using the *Tools/Filter* menu (***Appendix 3—figure 6***). Then we copy gene symbols from the filtered rows by doing a right click on any value in the *Gene symbol* column and selecting the option *Copy selected values from Gene symbol column* in the appeared pop-up menu. These genes then can be pasted into a text file to be used for the next stage.

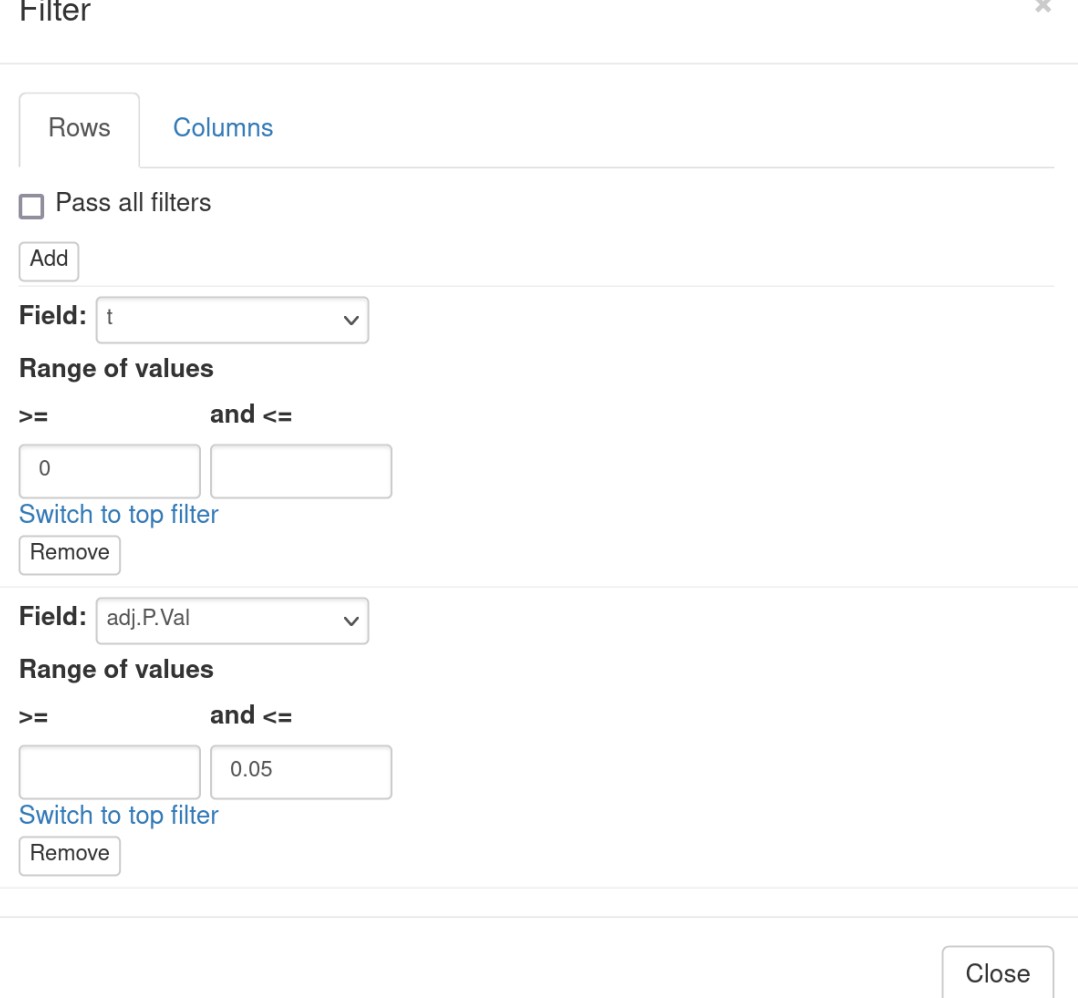

**Appendix 3—figure 6.** Filter tool settings for the *Id2*-dependent gene signature.

## *Rroid*-mediated regulation in group 1 ILCs

Mowel and colleagues suggested that the discovered *Rroid* locus regulation of ILCs development happens via regulation of *Id2* gene. In order to computationally confirm this hypothesis, the authors examined the enrichment of the previously obtained *Id2* gene signature in the dataset with *Rroid*-deficient and WT samples. The *Rroid* dataset generated by the authors is available in the GEO database under accession number GSE101458.

Once the dataset GSE101458 is loaded into Phantasus, we can generate a PCA plot as we did for the dataset GSE76466. All samples will be used; however, preliminary filtration of lowly expressed genes and normalization is still required. As in the previous section, it can be done with the *Tools/ Advanced normalization/Voom* menu. Alternatively, a reasonable result could be obtained by the following steps. Filter top 12,000 genes based on the row mean values: create the mean column using the *Tools/Create Calculated Annotation* menu and then use the top filter on it with *Tools/ Filter*. In the *Tools/Adjust* menu, check *One plus log 2* and *Quantile normalize* options to complete all preparations. Finally, build the PCA plot using the adjusted heatmap.

The PCA plot (*Appendix 3—figure 7*) indicates that the first samples in each of the two groups differ significantly from the rest, which could indicate that they are outliers. We will proceed with all of the samples but the results should be taken cautiously.

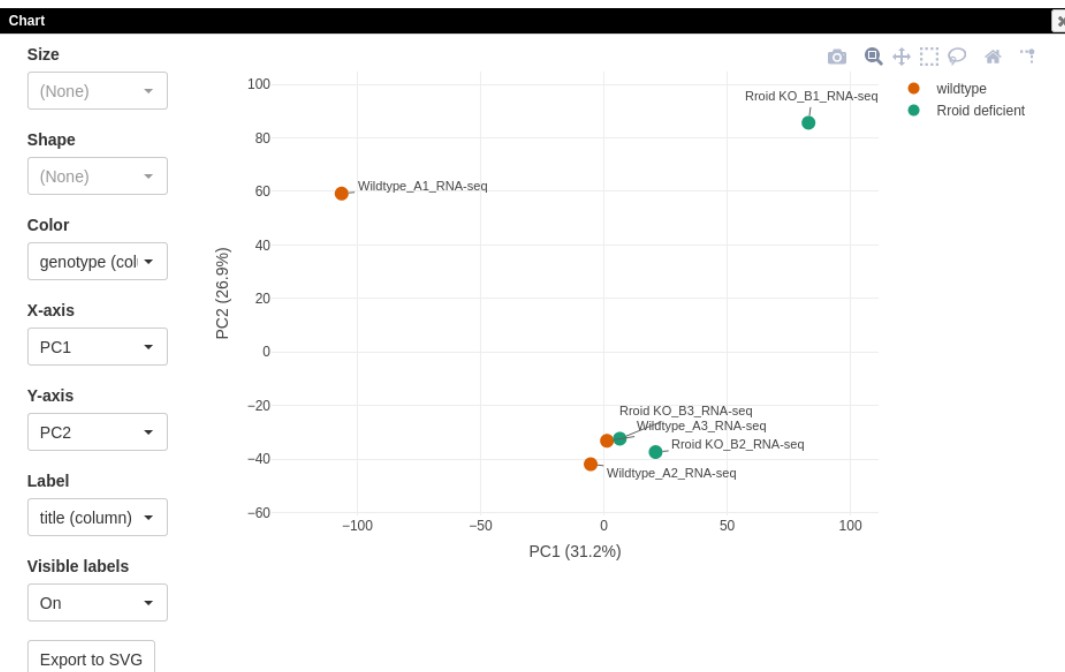

**Appendix 3—figure 7.** Principal component analysis (PCA) plot for dataset GSE101458.

Following the authors, we will use the DESeq2 tool (*Love et al., 2014*) for differential gene expression analysis. Notice that DESeq2 requires raw count matrix with all of the genes as an input, so all of the further steps are performed with the original dataset. Switch the tab to the first one, remove all filters by clicking on the *Remove* button in the *Tools/Filter* menu, and then run the *Tool/ Differential Expression/DESeq2* tool. Use *genotype* field to set up the comparison of *Rroid* deficient versus WT samples (*Appendix 3—figure 8*).

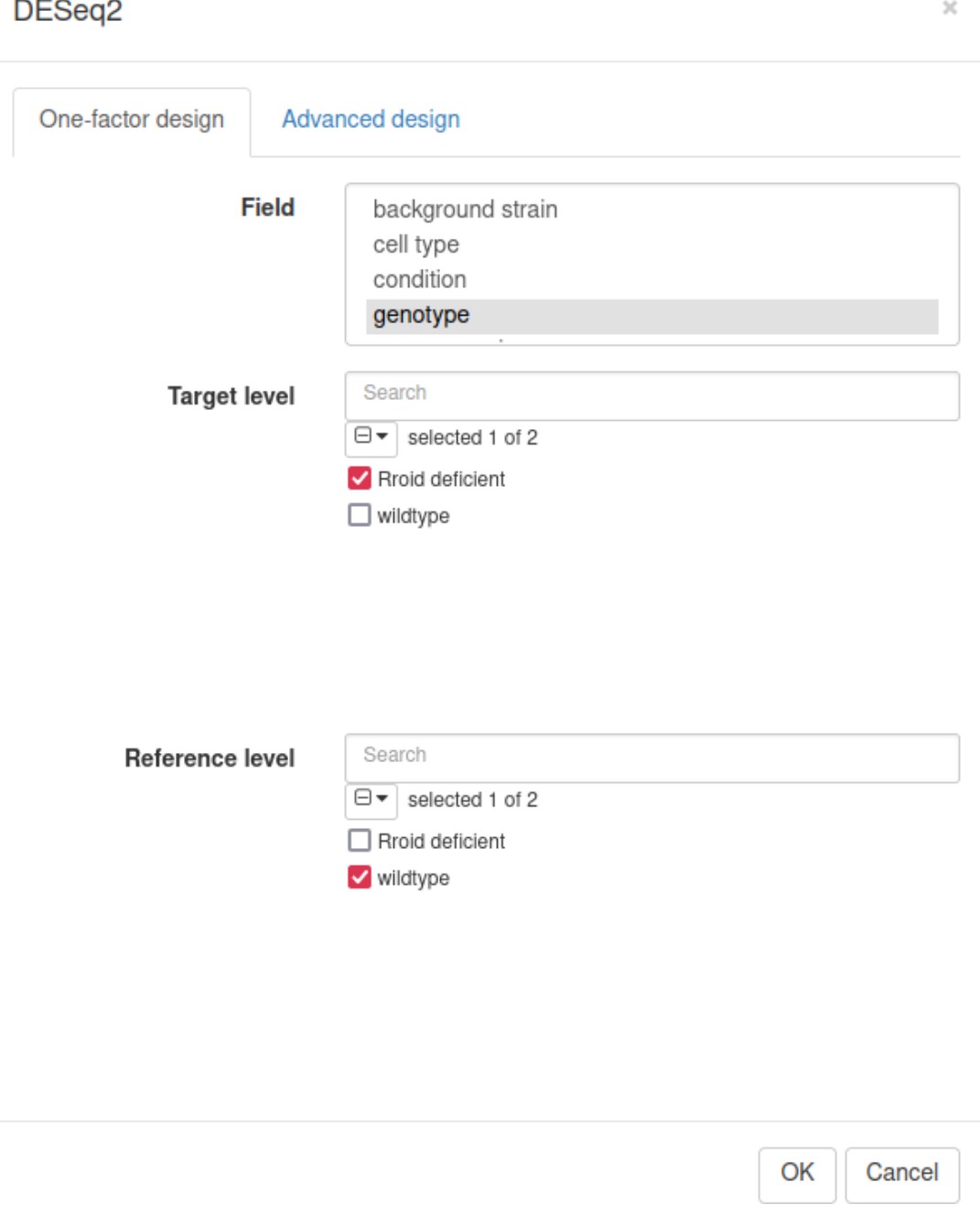

**Appendix 3—figure 8.** DESeq2 tool settings for the dataset GSE101458.

 After performing differential expression analysis, we can visualize it using the volcano plot. Click on the *stat* column three times to achieve the interleaved sorting of up- and downregulated genes, select several differentially expressed genes at the top of the table, and run *Tools/Plots/Volcano Plot* with the *Label by selected* option. The resulting volcano plot (*Appendix 3—figure 9*) reveals significant downregulation of *Id2* gene, which is consistent with the main hypothesis.

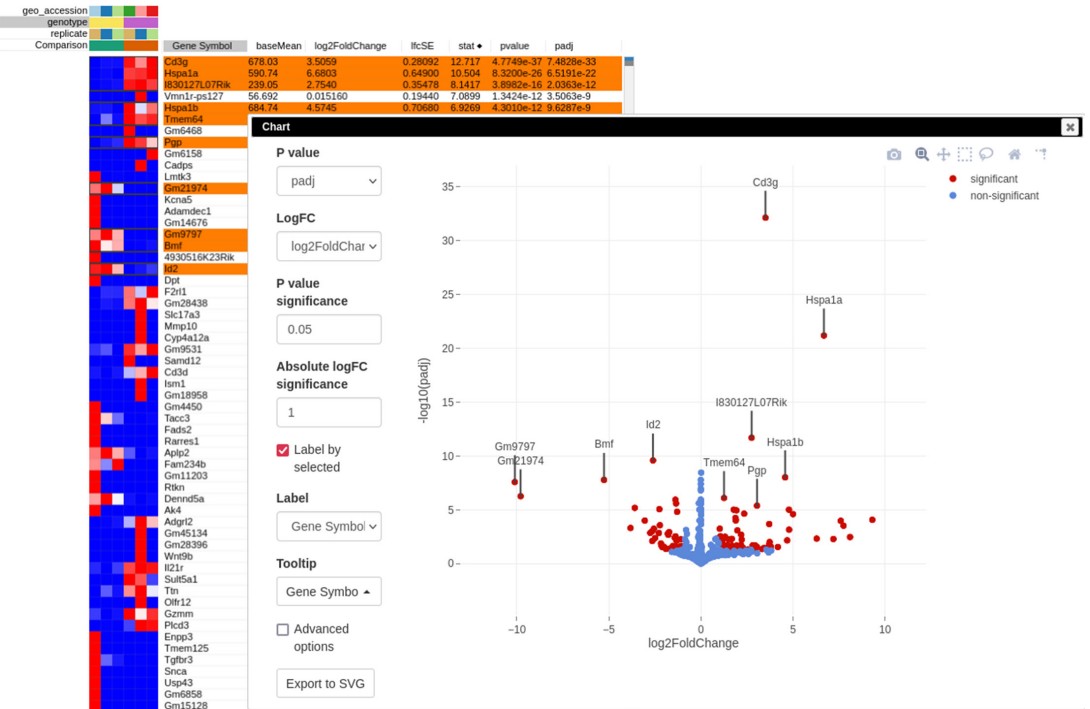

**Appendix 3—figure 9.** Volcano plot with differential gene expression results between wild-type (WT) and *Rroid*-deficient samples for dataset GSE101458.

Following *Mowel et al., 2017*, we aim to conduct the GSEA. GSEA requires to rank genes by the difference in the expression. Using the *stat* column values for that, we will build the enrichment plot for the *Id2*-dependent gene signature.

In order to get a cleaner heatmap representation of the gene expression matrix, we will use scaled values of the top 12,000 expressed genes. This can be reproduced in two steps. First, apply the top filter from the *Tools/Filter* tool on the *baseMean* column. Second, scale the filtered values using logarithmic (*One plus log2*) and quantile normalization options via *Tools/Adjust* menu.

Finally, to create a GSEA plot (*Appendix 3—figure 10*), we copy the previously saved gene signature (Section F), paste it into search field of the filtered DE results, and use the *Tools/Plots/GSEA Plot* menu. The resulting plot shows that the *Id2*-dependent genes are upregulated in the *Rroid*-deficient samples. This plot replicates figure 4E of the original paper by Mowel and colleagues.

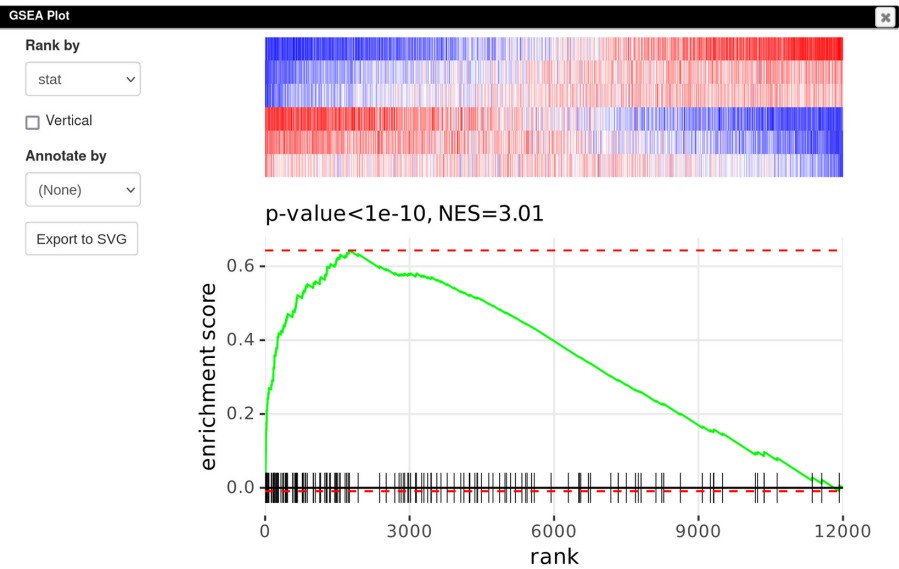

**Appendix 3—figure 10.** Enrichment plot of the *Id2*-dependent gene signature in dataset GSE101458.

