## [Editor Report]

This study introduces Phantasus, a useful tool accessible through both web and local applications, designed to analyze transcriptome data derived from microarray or RNA-seq technologies. The compelling tool facilitates normalization, data visualization, and differential expression analysis. Phantasus represents a valuable contribution to the biomedical community, enabling individuals without extensive bioinformatics expertise to analyze new transcriptomic data or reproduce studies effectively.

---

## [Decision Letter]

**Decision letter after peer review:**

Thank you for submitting your article "Phantasus: web-application for visual and interactive gene expression analysis" for consideration by *eLife*. Your article has been reviewed by 3 peer reviewers, including Marisa Nicolás as the Reviewing Editor and Reviewer #1, and the evaluation has been overseen by Aleksandra Walczak as the Senior Editor. The following individual involved in review of your submission has agreed to reveal their identity: Ivan Rodrigo Wolf (Reviewer #2).

Essential revisions (for the authors):

1) Improve English by checking and correcting misspelled words, such as typo errors, a few wrong or missing prepositions, punctuation, and the correct verb forms in some sentences.

2) Authors need to improve the details of local installation and tutorials for the end user.

3) Local functionality of the Phantasus through its Bioconductor package requires significant improvements to reach best practices in analyzing bulk RNA-seq data.

*Reviewer #1 (Recommendations for the authors):*

Phantasus is a good initiative as a tool that will be helpful for both experienced users in bioinformatics analysis and beginners. The manuscript describes all aspects of the tool needed to perform the tests with microarray samples and RNA-seq data.

For the case studies described in the manuscript, it would be a good idea for the authors to describe the RNA-seq tutorial in more detail.

The online version is user-friendly and aims to perform analyses using the tools from differential expression (by limma or DEGseq2), clustering (K-means, NN, or HC), Plots (Chart, PCA, GSEA, Volcano), and Pathway analysis (via Enrich or FGSEA).

The homepage contains a dataset selected by authors (GSE53986) and a list of diseases from TCGA (Acute Myeloid Leukemia to Uveal Melanoma). However, the information about the dataset for diseases of TCGA data needs to be included for the users.

Regarding the English written in the MS, using a spell checker, I found almost 60 misspelled words, such as typo errors. Also, the authors need to check a few wrong or missing prepositions, punctuation, and the right choice of verb forms in some sentences.

*Reviewer #2 (Recommendations for the authors):*

I would like to congratulate the authors for the development of Phantasus. The tool requires a small learning curve and can help researchers with less experience in bioinformatics. However, I have some general suggestions regarding the manuscript.

I notice some typos throughout the text, for your information:

PDF – page 8 line 181 – "procduces" – do you mean "produces"?

PDF – page 9 line 187 – "apthways" – do you mean "pathways"?

PDF – page 9 line 193 – there is no need to repeat the word "array" after "Arrays"

PDF – page 28 line 397 – do you mean "Submit" instead of "OK" ?

Also, the ARCHS4 is now plublished https://www.nature.com/articles/s41467-018-03751-6. I recommend changing the preprint citation to the final publication.

At the end of the manuscript, two appendices have tutorials on how to use the tool. However, the tutorial becomes superficial when using the RNA-Seq data in Appendix 3. I will cite some examples:

– It is not clear which column is used to sort the result table.

– There is no mention of the need to select the lines before opening the volcano graph to show the gene symbols on the graph.

– The padj and stat filter is mentioned only after the results are displayed.

– Contrast settings (such as what was selected in Class A and Class B in the Tool/Differential Expression/DESeq2 experimental menu entry) were not shown.

As such, I would like to see the same level of detail that is presented in the tutorial in Appendix 2, as RNA-Seq data is of great interest to researchers today.

Additionally, I believe that the authors need to improve the details of local installation and tutorials for the end user, below I will mention the main difficulties of the process.

When you open the Phantasus github page there are no instructions on how you can install Phantasus using the repository itself, only links to other sites and a redundant link to github repo itself. This can be improved with the installation information that is on Docker hub.

When visiting the image link on Docker Hub ther is no instructions on how to run or configure the container, but the instructions on how to install Phantasus R package through github. This can be improved with the information already present in the Phantasus documentation.

Now, regarding the R package installation instructions. I installed the listed dependencies and created an anaconda environment with a clean install of R. The Bioconductor package installation method does not work. Even the package available through the conda package installer did not work (but I believe this is the responsibility of the Bioconda maintainers and not the authors). The github install method (from the dockerhub page) didn't work either. Aware that installation errors can come from my computer's configuration, I chose to install using Docker.

The instructions on how to run the docker container in the documentation link on the github page let you download the container. However, the service only started correctly following the docker-compose instructions, probably due to some error in the docker run command from the documentation.

Finally, after these improvements in the installation documentation, the installation process can become less confusing, and the end user can benefit from Phantasus more easily.

---

## [Author Response]

Essential revisions (for the authors):1) Improve English by checking and correcting misspelled words, such as typo errors, a few wrong or missing prepositions, punctuation, and the correct verb forms in some sentences.2) Authors need to improve the details of local installation and tutorials for the end user.3) Local functionality of the Phantasus through its Bioconductor package requires significant improvements to reach best practices in analyzing bulk RNA-seq data.

First, we have revised the article, correcting misspellings and typos to ensure a clearer and more accurate text.

Next, we've revamped the package installation process. For local installation, an interactive setup procedure has been introduced, simplifying the creation of the necessary directory structure and file loading for complete functionality. Experienced users can opt for a server-like installation facilitated by the docker image and docker-compose script, providing a more streamlined configuration process. Furthermore, we implemented an HSDS-based storage that allows access to remote HDF5 files with RNA-seq count matrices, minimizing the size of required downloads.

We also have expanded the Phantasus functionality. We have updated and integrated new gene expression analysis tools. Notably, Voom and TMM normalizations have been added, along with the capacity to construct complex designs for Limma and DESeq2, contributing to a more robust and versatile application. We also expanded the list of the supported file formats.

Finally, due to the moving of the lab we have changed the official mirror of Phantasus to https://alserglab.wustl.edu/phantasus. The previous mirrors will continue to be maintained but will be gradually phased out in the future. In the process we’ve updated the RNA-seq counts databases to reflect the most recent versions of ARCHS4 and DEE2.

Reviewer #1 (Recommendations for the authors):Phantasus is a good initiative as a tool that will be helpful for both experienced users in bioinformatics analysis and beginners. The manuscript describes all aspects of the tool needed to perform the tests with microarray samples and RNA-seq data.

Thank you very much for your kind words and positive feedback on our article about Phantasus. We truly appreciate your thoughtful review and are delighted to hear that you find Phantasus to be a valuable tool.

For the case studies described in the manuscript, it would be a good idea for the authors to describe the RNA-seq tutorial in more detail.

Indeed, the previous tutorial on RNA-seq data analysis was not as comprehensive, especially after we added a number of more RNA-seq specific tools. Accordingly we've made a number of significant changes. The key update involved a rewrite of the gene signature acquisition section to showcase the voom/limma pipeline for RNA-seq differential expression analysis. Furthermore, we've included screenshots detailing the settings for differential expression in both parts, along with guidance on row filtration and count normalization. We hope that these additions improved the comprehensiveness and clarity of the tutorial.

The online version is user-friendly and aims to perform analyses using the tools from differential expression (by limma or DEGseq2), clustering (K-means, NN, or HC), Plots (Chart, PCA, GSEA, Volcano), and Pathway analysis (via Enrich or FGSEA).The homepage contains a dataset selected by authors (GSE53986) and a list of diseases from TCGA (Acute Myeloid Leukemia to Uveal Melanoma). However, the information about the dataset for diseases of TCGA data needs to be included for the users.

We are glad to hear that you find our application user-friendly. However, indeed, a proper description of TCGA datasets was lacking from the homepage. To address this concern, we have updated the homepage, adding more information on how the datasets were obtained and incorporating direct links to detailed descriptions of the TCGA datasets.

Regarding the English written in the manuscript, using a spell checker, I found almost 60 misspelled words, such as typo errors. Also, the authors need to check a few wrong or missing prepositions, punctuation, and the right choice of verb forms in some sentences.

Thank you for highlighting the language and grammar issues in our manuscript. We have conducted additional spelling and grammar checks and corrected the errors.

Reviewer #2 (Recommendations for the authors):I would like to congratulate the authors for the development of Phantasus. The tool requires a small learning curve and can help researchers with less experience in bioinformatics. However, I have some general suggestions regarding the manuscript.

Thank you very much for your kind words and appreciation of our tool. We do hope that the changes made during this revision would improve its usefulness even further.

I notice some typos throughout the text, for your information:PDF – page 8 line 181 – "procduces" – do you mean "produces"?PDF – page 9 line 187 – "apthways" – do you mean "pathways"?PDF – page 9 line 193 – there is no need to repeat the word "array" after "Arrays"PDF – page 28 line 397 – do you mean "Submit" instead of "OK" ?

Thank you for highlighting the language and grammar issues in our manuscript. We have conducted additional spelling and grammar checks and corrected the errors.

Also, the ARCHS4 is now plublished https://www.nature.com/articles/s41467-018-03751-6. I recommend changing the preprint citation to the final publication.

Thank you for noticing the incorrect citation. We have revised the citation to reflect the published version

At the end of the manuscript, two appendices have tutorials on how to use the tool. However, the tutorial becomes superficial when using the RNA-Seq data in Appendix 3. I will cite some examples:– It is not clear which column is used to sort the result table.– There is no mention of the need to select the lines before opening the volcano graph to show the gene symbols on the graph.– The padj and stat filter is mentioned only after the results are displayed.– Contrast settings (such as what was selected in Class A and Class B in the Tool/Differential Expression/DESeq2 experimental menu entry) were not shown.As such, I would like to see the same level of detail that is presented in the tutorial in Appendix 2, as RNA-Seq data is of great interest to researchers today.

Thank you for these comments, we acknowledge the limitations of the previous tutorial on RNA-seq data analysis, particularly in providing detailed guidance. In response, we have made significant revisions to enrich the description and enhance user understanding. Specifically, we have refined the gene signature section to highlight the newly implemented voom/limma pipeline for RNA-seq differential expression analysis. We have also included screenshots of settings for differential expression analysis (Appendix 3—figures 4 and 8), and provided additional guidance on row filtration (Appendix 3—figure 6) and count normalization (Appendix 3—figure 2). We hope that these adjustments will address the specific concerns you raised and help us to create a more thorough and explicit tutorial.

Additionally, I believe that the authors need to improve the details of local installation and tutorials for the end user, below I will mention the main difficulties of the process.When you open the Phantasus github page there are no instructions on how you can install Phantasus using the repository itself, only links to other sites and a redundant link to github repo itself. This can be improved with the installation information that is on Docker hub.When visiting the image link on Docker Hub ther is no instructions on how to run or configure the container, but the instructions on how to install Phantasus R package through github. This can be improved with the information already present in the Phantasus documentation.Now, regarding the R package installation instructions. I installed the listed dependencies and created an anaconda environment with a clean install of R. The Bioconductor package installation method does not work. Even the package available through the conda package installer did not work (but I believe this is the responsibility of the Bioconda maintainers and not the authors). The github install method (from the dockerhub page) didn't work either. Aware that installation errors can come from my computer's configuration, I chose to install using Docker.The instructions on how to run the docker container in the documentation link on the github page let you download the container. However, the service only started correctly following the docker-compose instructions, probably due to some error in the docker run command from the documentation.Finally, after these improvements in the installation documentation, the installation process can become less confusing, and the end user can benefit from Phantasus more easily.

Thank you for such detailed feedback on installation procedures. Taking into account your experience we have largely revised our installation instructions as well as the installation procedure itself.

First of all, we restructured R package installation. Newly added ‘setupPhantasus’ function is able to create all necessary configuration files and provides an interactive dialog with the user that helps to load all necessary data files from our official cache mirror (https://alserglab.wustl.edu/files/phantasus/minimal-cache/). These files unlock features of Phantasus which were previously unavailable without additional setup such as gene annotation mapping and gene set enrichment analysis. Moreover, we have set up an HSDS server to store HDF5 files with the precomputed gene counts and developed a helper R package phantasusLite (https://bioconductor.org/packages/phantasusLite) for accessing these data from R. Together it allows an easy remote access to gene counts for the specified dataset, without the need to download and store large HDF5 files. Thus, with help of the new installation procedure locally installed Phantasus now has the whole functionality available at the official mirror. Docker-based installation follows the same approach.

With the updated installation procedure and based on your comments we have carefully revised installation documentation. The most comprehensive installation description is available at https://ctlab.github.io/phantasus-doc/installation. For GitHub and DockerHub we have added short quick starts for R- and Docker- based installation respectively, with the links to the more detailed description. We believe that this approach provides balance between ease of access and reduced redundancy and simplified maintanance of instructions.

We hope that local installation now becomes more smooth and well defined.